# PROVABLY LEARNING CONCEPTS BY COMPARISON

## ABSTRACT

We are born with the ability to learn concepts by comparing diverse observations. This helps us to understand the new world in a compositional manner and facilitates extrapolation, as objects naturally consist of multiple concepts. In this work, we argue that the cognitive mechanism of comparison, fundamental to human learning, is also vital for machines to recover true concepts underlying the data. This offers correctness guarantees for the field of concept learning, which, despite its impressive empirical successes, still lacks general theoretical support. Specifically, we aim to develop a theoretical framework for the identifiability of concepts with multiple classes of observations. We show that with sufficient diversity across classes, hidden concepts can be identified without assuming specific concept types, functional relations, or parametric generative models. Interestingly, even when conditions are not globally satisfied, we can still provide alternative guarantees for as many concepts as possible based on local comparisons, thereby extending the applicability of our theory to more flexible scenarios. Moreover, the hidden structure between classes and concepts can also be identified nonparametrically. We validate our theoretical results in both synthetic and real-world settings.

## 1 INTRODUCTION

Humans possess an innate ability to learn concepts by comparing diverse classes of observations, a process foundational to cognitive development (Rosch, 1973; Fodor & Pylyshyn, 1988). For example, a child distinguishes between different types of animals not by memorizing each species separately, but by observing and comparing differences between various species, thereby identifying the unique concepts that define each group (e.g., Fig. 1). This mechanism of learning through comparison has been extensively studied and verified across various fields, including psychology and neuroscience, affirming its universality and effectiveness (Bruner et al., 1957).

Meanwhile, in machine learning, the extraction of conceptual features is crucial for the development of robust and interpretable models, illustrating the integration of cognitive principles into machine intelligence (Valiant, 1984; Mitchell, 1997). Recent research has achieved notable success in deriving human-interpretable concepts from various data modalities with different formulations of the problem (Bau et al., 2017; Radford et al., 2017; Alvarez Melis & Jaakkola, 2018; Kim et al., 2018; Zhou et al., 2018; Yeh et al., 2020; Koh et al., 2020; Du et al., 2021; Bai et al., 2022; Achtibat et al., 2022; Crabbé & van der Schaar, 2022; Liu et al., 2023; Park et al., 2023; Jiang et al., 2024). These concepts have proven beneficial for tasks such as extrapolation (Janner et al., 2022; Lachapelle et al., 2023; Du & Kaelbling, 2024), explanation (Alvarez Melis & Jaakkola, 2018; Sreedharan et al., 2020; Leemann et al., 2023; Poeta et al., 2023), and decision-making (Grupen et al., 2022; Zabounidis et al., 2023; Delfosse et al., 2024). Furthermore, advancements in this domain have significantly contributed to scientific discovery, particularly in healthcare (Clough et al., 2019; Jia et al., 2022).

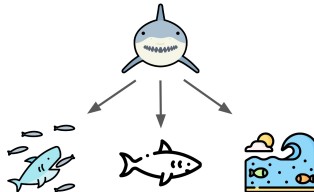

Figure 1: The class "shark" has concepts like "predator," "sleek body," and "ocean."

While numerous methods have been developed to extract concepts from data, most provide only empirical support and lack theoretical guarantees concerning the correctness of the recovered concepts. With the help of specific parametric assumptions, few studies have explored the identifiability of concept learning. For example, by assuming all concepts are linearly related, recent research (Rajendran et al., 2024) has shown that the concept space can be identified up to a linear transformation.

Another line of research has tackled object-centric learning, attempting to identify individual objects as groups of pixels (slots), such as trees or dogs, while excluding more abstract concepts like lighting and styles. In addition to these concept type restrictions, further assumptions are also required for the identifiability results, such as no occlusion between objects (Brady et al., 2023; Wiedemer et al., 2024) or the additivity of the generating process (Lachapelle et al., 2023; Wiedemer et al., 2024). These studies mark significant exploration toward understanding concept learning. At the same time, the constraints imposed on concept types and functional relationships may limit the confidence to fully account for the empirical success observed in concept learning from real-world scenarios. Therefore, despite significant empirical progress, a fundamental question in concept learning remains unanswered:

*In the most general cases, which concepts can we reliably recover?*

We try to provide an answer by drawing inspiration from the fundamental cognitive mechanism through which humans learn concepts, i.e., comparing diverse classes of observations. For an infant, devoid of empirical world knowledge, it is impossible to learn new concepts from two classes of observations if they share an identical set of concepts. It is only through discerning the differences between these classes that humans can unravel and understand previously unseen concepts. As a result, in the most general setting, the essential information for provably learning hidden concepts must pertain to the diversity present among different classes.

Inspired by this cognitive process of learning by comparison, we establish a set of theoretical guarantees on concept learning in the general setting. We show that hidden concepts can be identified without relying on assumptions about the nature of the concepts or specific parametric models, provided there is sufficient diversity across classes. Specifically, we first prove that for any pair of classes, the unique part of the concepts for each class can be disentangled from the remaining concepts (Thm. 1). This pairwise comparison[1] serves as a foundational prototype for learning concepts, enabling the flexible identifiability of as many concepts as possible, given that they exhibit enough diversity, even when others do not. We then extend the pair-wise identifiability to learn unique concepts from an arbitrary subset of classes (Prop. 1). Given that most related works rely on global assumptions for all concepts and fail to offer guarantees when assumptions are partially violated for some concepts, the proposed flexible identifiability by local comparisons provides unique practical value, since real-world scenarios often do not perfectly conform to ideal conditions for all concepts.

Furthermore, with sufficient diversity across different classes of observations, we prove the nonparametric identifiability for all class-related hidden concepts up to an element-wise transformation and permutation (Thm. 2). For other invariant background concepts, such as "chromatic" that remain consistent across all classes, we can also identify them under appropriate structural diversity conditions (Prop. 2). Consequently, we introduce, to the best of our knowledge, one of the first frameworks for concept identifiability in the general setting that does not confine itself to specific concept types or parametric generative models. Moreover, the connective structure between classes and concepts can also be recovered in a nonparametric way (Prop. 3). Our theoretical results are substantiated through empirical validation on synthetic data and four different real-world datasets.

## 2 PRELIMINARIES

In this section, we introduce the problem setting as well as some essential notations. Fig. 2 illustrates the key notations and relations of the considered setting. We also provide a structured summary of notations in Appx. A for a quick reference.

**Data-generating Process.** Let $\mathbf{x} = (\mathbf{x}_1, \ldots, \mathbf{x}_m) \in \mathcal{X} \subseteq \mathbb{R}^m$ be a vector representing observed variables. We assume that the observation $\mathbf{x}$ is generated by hidden *concepts* $\mathbf{z} = (\mathbf{z}_A, \mathbf{z}_B) \in \mathcal{Z} \subseteq \mathbb{R}^n$. The generating process is as follows:

$$\mathbf{x} := f(\mathbf{z}), \qquad (1)$$

where we divide $\mathbf{z}$ into the class-dependent part $\mathbf{z}_A = (\mathbf{z}_1, \ldots, \mathbf{z}_{n_A}) \in \mathcal{Z}_A \subseteq \mathbb{R}^{n_A}$ and class-independent part $\mathbf{z}_B =$

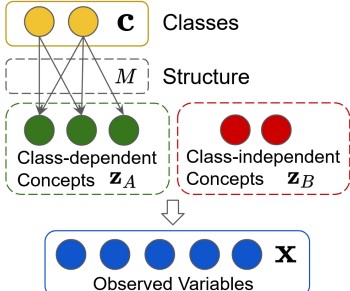

Figure 2: The problem setting.

---

[1]It might be worth noting that learning by comparison serves as an inspiration for our identifiability theory, rather than being a specific estimation method like contrastive learning.

$(\mathbf{z}_{n_A+1}, \ldots, \mathbf{z}_n) \in \mathcal{Z}_B \subseteq \mathbb{R}^{n_B}$. The class-dependent part $\mathbf{z}_A$ and class-independent part $\mathbf{z}_B$ are conditionally independent given observed *classes* $\mathbf{c} = (\mathbf{c}_1, \ldots, \mathbf{c}_u) \subseteq \mathbb{R}^u$, i.e., $p(\mathbf{z}|\mathbf{c}) = p(\mathbf{z}_A|\mathbf{c})p(\mathbf{z}_B)$. We denote the number of classes as $k$. The density $p(\mathbf{z}|\mathbf{c})$ is smooth and positive. Since $\mathbf{z}_A$ depends on the classes $\mathbf{c}$, we represent $\mathbf{z}_A := g(\mathbf{c}, \theta)$, where $\theta$ denotes a set of other factors including potential noise. Let $A_i$ denote the index set of concepts corresponding to class $\mathbf{c}_i$, with the associated concepts represented as $\mathbf{z}_{A_i}$. Likewise, $\mathbf{z}_{A_i \setminus A_j}$ refers to the difference in the concept sets between classes $\mathbf{c}_i$ and $\mathbf{c}_j$. The generating function $f$ is a general injective function that encodes potentially complex mixing procedures to generate the observational data. Meanwhile, we do not constrain $\mathbf{z}$ to be of specific distributions like Gaussian. Consequently, we consider a general formulation of the problem that covers different types of concepts and nonparametric generative models. Here is a real-world example of how the data-generating process may be instantiated:

**Example 1.** *Consider images of animals in an aquarium, where the observed variables $\mathbf{x}$ represent image pixels. The different animal types (e.g., "shark" and "turtle") correspond to classes c. Class-dependent concepts might include attributes like "predator," "sleek body," and "ocean" (see, e.g., Fig. 1), while class-independent concepts could be "lighting" and "position." The hidden generative process of each image depends on all of these concepts, though only some are specific to each class.*

**Technical Notations.** Throughout this work, for any matrix $S$, we use $S_{i,:}$ to denote its $i$-th row, and $S_{:,j}$ to denote its $j$-th column. For any set of indices $\mathcal{I} \subset \{1, \ldots, m\} \times \{1, \ldots, n\}$, analogously, we have $\mathcal{I}_{i,:} := \{j \mid (i,j) \in \mathcal{I}\}$ and $\mathcal{I}_{:,j} := \{i \mid (i,j) \in \mathcal{I}\}$. We also denote the support of the matrix $S \in \mathbb{R}^{a \times b}$ as $\mathrm{supp}(S) := \{(i,j) \mid S_{i,j} \neq 0\}$. With a slight abuse of notation, we reuse $\mathrm{supp}(\cdot)$ to denote the support of a matrix-valued function $\mathbf{S}(\Theta) : \Theta \to \mathbb{R}^{a \times b}$, i.e., $\mathrm{supp}(\mathbf{S}(\Theta)) := \{(i,j) \mid \exists \theta \in \Theta, \mathbf{S}(\theta)_{i,j} \neq 0\}$. Then we define $\mathcal{D}$ as the support of $D_\mathbf{c} g$, i.e., $\mathcal{D} = \mathrm{supp}(D_\mathbf{c} g)$, where $D_\mathbf{c} g$ represents the partial derivative of $g$ w.r.t. $\mathbf{c}$. Moreover, we define $\mathcal{T}$ as a set of matrices with the same support of $\mathbf{T}$ in $D_{\hat{\mathbf{c}}} \hat{g} = \mathbf{T} D_\mathbf{c} g$, where $\mathbf{T}$ is a matrix-valued function. In addition, given a subset $\mathcal{S} \subseteq \{1, \ldots, n\}$, the subspace $\mathbb{R}^n_\mathcal{S}$ is defined as:

$$\mathbb{R}^n_\mathcal{S} := \{s \in \mathbb{R}^n \mid s_i = 0 \text{ if } i \notin \mathcal{S}\}, \tag{2}$$

where $s_i$ is the $i$-th element of the vector $s$. Throughout the work, we use the hat symbol (e.g., $\hat{\mathbf{z}}$) to denote estimated quantities, such as $\hat{\mathbf{z}}$ for estimated concepts. Since the considered problem is identifiability, the theory is agnostic to estimators and the goal is to fit the marginal distribution $p(\mathbf{x})$ with model (learner) $\hat{f}$ and estimated variables $\hat{\mathbf{z}}$ to achieve certain identifiability. We introduce several identifiability objectives (Hyvärinen & Morioka, 2017; Lachapelle et al., 2022; Zheng et al., 2022; Kong et al., 2022; Hyvärinen et al., 2024) that are common in the literature as follows:

**Definition 1 (Element-wise Identifiable).** *The set of latent variables $\mathbf{z} \subseteq \mathbb{R}^n$ are element-wise identifiable if there exists an invertible function $h_i : \mathbb{R} \to \mathbb{R}$ and a permutation $\pi$ s.t. $\hat{\mathbf{z}}_i = h_i(\mathbf{z}_{\pi(i)})$.*

**Definition 2 (Subspace-wise Identifiable).** *The set of latent variables $\mathbf{z} \subseteq \mathbb{R}^n$ are subspace-wise identifiable if there exists an invertible function $h : \mathbb{R}^n \to \mathbb{R}^n$ s.t. $\hat{\mathbf{z}} = h(\mathbf{z})$.*

It might be worth noting that the subspace-wise identifiability implies the disentanglement between subsets of latent variables. For instance, if $\mathbf{z}_B$ is subspace-wise identifiable, then $\mathbf{z}_B$ will not contain any information from $\mathbf{z}_A$ after estimation. The subspace-wise identifiability is commonly used in the literature (Von Kügelgen et al., 2021; Kong et al., 2022; Li et al., 2024; Yao et al., 2024).

**Connective Structure.** Based on these, we define the *structure* $M$ as a binary matrix with the support $\mathcal{D}_{:n_A,:}$. The class-dependent part $\mathbf{z}_A$ can be further represented as

$$p(\mathbf{z}_A|\mathbf{c}) = \prod_{i=1}^{n_A} p(\mathbf{z}_i | M_{i,:} \odot \mathbf{c}), \tag{3}$$

where $M_{i,:}$ is the $i$-th row of $M$. The operator $\odot$ denotes the element-wise (Hadamard) product. Since classes $\mathbf{c}$ are not connected to class-independent part $\mathbf{z}_B$, $M$ illustrates the connective structure between classes $\mathbf{c}$ and concepts $\mathbf{z}$. The conditional independence provides a form of modularity commonly adopted in prior work on identifiable latent variable models (Hyvärinen & Morioka, 2016; Khemakhem et al., 2020a; Sorrenson et al., 2020; Lachapelle et al., 2022; Hyvärinen et al., 2024). It may be particularly natural in our class-concept framework; for example, while the concepts "wings" and "feathers" are related, they become conditionally independent given the class variable "bird."

## 3 IDENTIFIABILITY THEORY

Without any assumptions on specific concept types, functional relations, or parametric generative models, to what extent can we provably learn hidden concepts from diverse classes of observations?

To answer this, in Section 3.1, we first prove that the unique concepts in any pair of classes can be disentangled from the remaining ones (Thm. 1). Based on this, we can fully leverage the diversity in the data and provide flexible identifiability for any subset of concepts, as long as there exists sufficient diversity for local comparison (Prop. 1). For the global identification, in Section 3.2, we prove the nonparametric identifiability for all class-dependent hidden concepts (Thm. 2) under the structural diversity condition (Assump. 1). Together with a sparsity condition for the remaining class-independent part, all hidden concepts can be identified up to trivial indeterminacy (Prop. 2). Furthermore, in Section 3.3, we show that we can also recover the hidden connective structure between classes and concepts (Prop. 3), providing further insights into the latent compositional relations.

### 3.1 LEARNING CONCEPTS BY LOCAL COMPARISON

Humans learn concepts by leveraging the diversity across classes. We argue that the fundamental mechanism in this cognitive process is learning through pair-wise comparison, since any two classes can only be distinguished by identifying their unique concepts. Pairwise comparison thus serves as the basic unit for concept learning across multiple classes, as comparisons among any set of classes can be reduced to pairs. In the following theorem, we prove that the unique concepts between any pair of classes can be disentangled from the remaining concepts, of which the proof is in Appx. B.1.

**Theorem 1.** *Let the observed data be a sufficiently large sample generated by a model defined in Sec. 2. Suppose for each $i \in \{1, \ldots, n_A\}$, there exist a set of points $\{(c, \theta)^{(\ell)}\}_{\ell=1}^{|\mathcal{D}_{:,i}|}$, a point $(c, \theta)^{(r)}$, and a matrix $T \in \mathcal{T}$ such that the following conditions hold:*

> *i. The Jacobian spans its support space, i.e., $\mathrm{span}\{D_{\mathbf{c}}g((c, \theta)^{(\ell)})_{:,i}\}_{\ell=1}^{|\mathcal{D}_{:,i}|} = \mathbb{R}^{n_A}_{\mathcal{D}_{:,i}}$, and $\left[ T D_{\mathbf{c}}g((\mathbf{c}, \theta)^{(\ell)}) \right]_{:,i} \in \mathbb{R}^{n_A}_{\hat{\mathcal{D}}_{:,i}}$.*
>
> *ii. The Jacobian $D_{\mathbf{c}}g((\mathbf{c}, \theta)^{(r)})$ is of full row rank.*

*Then for any two classes $\mathbf{c}_i$ and $\mathbf{c}_j$, there exists a permutation $\pi$ such that $\hat{\mathbf{z}}_{\pi(A_i \setminus A_j)}$, do not depend on the latent concepts $\mathbf{z}_{A_j}$ associated with class $\mathbf{c}_j$, and $\hat{\mathbf{z}}_{\pi(A_j \setminus A_i)}$ do not depend on the latent concepts $\mathbf{z}_{A_i}$ associated with class $\mathbf{c}_i$.*

Theorem 1 demonstrates the process of learning through pair-wise comparison, which is fundamental to the learning mechanism. It is worth noting that the identifiability theory remains agnostic to the choice of estimator, provided the marginal distributions of the observations are matched. The results demonstrate that for any pair of classes, the unique concepts specific to each class can be disentangled from the other concepts. Additionally, we extend the theoretical guarantees of pairwise comparisons to arbitrary class sets, facilitating more efficient learning in complex scenarios:

**Proposition 1.** *Let the observed data be a sufficiently large sample generated by a model defined in Sec. 2. Suppose that the assumptions in Thm. 1 hold. Then, for a set of classes $\mathbf{c}_I$ and its corresponding concept sets $\mathbf{z}_{A_I}$ with a set of indices $I$, there exists a permutation $\pi$ that the unique part of a concept set for the class $\mathbf{c}_i$, i.e., $\hat{\mathbf{z}}_{\pi(A_i \setminus A_{I \setminus i})}$, does not depend on the latent concepts associated with other classes, i.e., $\mathbf{z}_{A_{I \setminus i}}$.*

> **Insights.** Theorem 1 and Proposition 1 show that as long as there exists any diversity between different classes, we can identify the corresponding hidden concepts with theoretical guarantees. This aligns with the fundamental cognitive mechanism of learning and offers a more flexible method to locally exploit available information. In contrast, most prior identifiability conditions focus on the entire system, often losing guarantees if any part violates the assumptions.

**Discussion on Assumptions.** The assumption here helps ensure the connection between the dependency structure and the Jacobian of the function in the general nonlinear cases, following the similar spirit in (Lachapelle et al., 2022; Zheng et al., 2022). In general, it avoids pathological cases where all samples originate from highly restricted sub-populations that only cover a degenerate

subspace. The first part makes sure that there are at least $|\mathcal{D}_{:n_A,i}|$ data points such that the Jacobian function spans the support space, which is almost guaranteed asymptotically. The condition $\left[\mathrm{T}D_{\mathbf{c}}g((\mathbf{c},\theta)^{(\ell)})\right]_{:,i} \in \mathbb{R}^{n_A}_{\hat{\mathcal{D}}_{:,i}}$ is also mild since $\hat{\mathcal{D}}_{:,i} = \mathbf{T}D_{\mathbf{c}}g((\mathbf{c},\theta)^{(\ell)})$ always resides in $\mathbb{R}^{n_A}_{\hat{\mathcal{D}}_{:,i}}$. Even in some rare cases where the matrix does not fit the support due to some generic combination of values, the assumption is still almost always satisfied asymptotically. This is because it only necessitates the existence of one matrix in the entire space ($\mathrm{T} \in \mathcal{T}$, where $\mathcal{T}$ denotes a set of matrices with the same support of $\mathbf{T}$). The second part avoids rank-deficiency and has been extensively employed in the literature (Hyvärinen et al., 2024). An illustrative example is as follows:

**Example 2.** *Suppose there exist two samples with their corresponding Jacobians given by $D_{\mathbf{c}}g((c,\theta)^{(1)})_{:,i} = (0,1,2)$ and $D_{\mathbf{c}}g((c,\theta)^{(2)})_{:,i} = (0,3,4)$. Clearly, these two vectors span a 2-dimensional subspace. We can also find a matrix $\mathrm{T}$ (e.g., a binary matrix with the same support as $\mathbf{T}$) s.t. $\left[\mathrm{T}D_{\mathbf{c}}g((c,\theta)^{(\ell)})\right]_{:,i} \in \mathbb{R}^{n_A}_{\hat{\mathcal{D}}_{:,i}}$ for $\ell \in \{1,2\}$. Any invertible function satisfies the full rank condition. Since identifiability theory considers an infinite number of samples, the requirement for several non-degenerate samples is almost always satisfied asymptotically.*

**Implications.** Theorem 1 demonstrates that for any given pair of classes and their corresponding sets of hidden concepts, the unique concepts in each class can be disentangled from all the remaining concepts. This process is fundamental to the cognitive mechanism of learning through comparison. Consider an infant with no prior experience of the world: when presented with two classes, such as a cat and a dog, the infant learns and memorizes the unique concepts associated with each class, such as "meows" for the cat and "barks" for the dog. The invariant concepts, like "furry" or "four-legged," cannot be distinctly learned because they do not provide distinguishing information between the classes. From a cognitive science perspective, infants and young learners rely heavily on contrastive features to form distinct categories and concepts (Eimas et al., 1971). For instance, if an infant repeatedly hears a cat meow and a dog bark, they begin to associate these unique sounds with the respective animals. In contrast, shared attributes like fur or four legs do not stand out because they do not help in differentiating between the two animals. This emphasizes the role of unique concepts in early learning and memory, highlighting how pair-wise comparisons are essential in the process of discovering the new world. For machines to learn without prior knowledge, we argue that similar mechanisms also help.

Proposition 1 extends these theoretical guarantees from pair-wise comparisons to local comparisons among multiple classes. Although pair-wise comparison is fundamental to the learning mechanism, local comparison is more efficient in complex scenarios. For instance, when an infant is exposed to a variety of stimuli, they do not learn by isolating pairs indefinitely. Instead, they begin to discern patterns and unique features within a broader context, comparing multiple classes simultaneously. For example, a child distinguishing between a cat, a dog, and a bird must identify unique concepts such as "meows," "barks," and "chirp." As the child interacts with these animals in different contexts—perhaps hearing a bird chirp in the park, a dog bark at home, and a cat meow in the neighbor's yard—they learn to associate specific sounds and behaviors with each animal. This local comparison ensures that even as the number of classes increases, the child can efficiently disentangle and learn the unique concepts of each class, providing a more complete understanding of the new environment.

Besides being the foundation for the learning process, the principles of local comparisons in both Thm. 1 and Prop. 1 also enable partial identifiability for a subset of concepts when diversity is not universally satisfied across all classes and concepts. Previous theoretical studies on concept learning often assume that certain conditions, such as linearity or additivity, apply universally to all concepts. While these assumptions can simplify the conceptual space and the generating process, they can not offer any guarantees for any concepts when there exists any degree of violation. However, since real-world scenarios are often complex and unpredictable, it is relatively rare for these assumptions to hold true universally. Most latent variable identifiability works also face the same challenge dealing with partial assumption violation (Zheng et al., 2022; Kong et al., 2022; Zheng & Zhang, 2023; Hyvärinen et al., 2024). Unlike our local or even pair-wise identification strategy, these methods lack the flexibility to recover arbitrary parts of the hidden process in a localized manner. Fortunately, with the proposed theory based on local comparisons (Thm. 1 and Prop. 1), we can leverage the diversity in observations to recover the hidden system as much as possible, even when the degree of diversity does not support global identifiability. For instance, in scenarios where some classes are very similar and several concepts are shared across all classes, these concepts cannot be learned through comparison. However, we can still achieve appropriate identifiability for the other concepts

with sufficient diversity. Notably, these flexible guarantees do not come with the cost of more restrictive conditions—the identifiability theory still applies to most generating processes without assumptions on specific concept types, functional relations, or parametric generative models.

## 3.2 LEARNING CONCEPTS BY GLOBAL COMPARISON

Inspired by the mechanism of local comparison, we have shown that it is possible to fully leverage the diversity among different classes of observations to recover hidden concepts as much as possible. This naturally leads us to consider the conditions required for identifying all hidden concepts in a global manner. We first prove that, under the condition of *Structural Diversity* (Assump. 1), all class-dependent concepts are identifiable up to a composition of a permutation and an element-wise invertible transformation (Thm. 2). The proof is included in Appx. B.3.

**Assumption 1.** *(Structural Diversity) For any class-dependent concept $\mathbf{z}_i$, there exists a set of indices $J$ ($|J| > 1$) and $j \in J$ where $M_{i,j} \neq 0$ and $M_{i,k} = 0$ for all $k \in J$, $k \neq j$, and $M_{i,J\setminus\{j\}}$ is the only row with all zero entries in $M_{:,J\setminus\{j\}}$.*

**Theorem 2.** *Let the observed data be a sufficiently large sample generated by a model defined in Sec. 2. In addition to the assumptions in Thm. 1 and Assump. 1, suppose for any set $A_{\mathbf{z}} \subseteq \mathcal{Z}$ with non-zero probability measure and cannot be expressed as $B_{\mathbf{z}_B} \times \mathbf{z}_A$ for any $B_{\mathbf{z}_B} \subset \mathcal{Z}_B$, there exist two values of $\mathbf{c}$, i.e., $c^{(k)}$ and $c^{(v)}$ (which may vary across different $A_{\mathbf{z}}$), that*

$$\int_{\mathbf{z} \in A_{\mathbf{z}}} p(\mathbf{z} \mid c^{(k)}) d\mathbf{z} \neq \int_{\mathbf{z} \in A_{\mathbf{z}}} p(\mathbf{z} \mid c^{(v)}) d\mathbf{z}.$$

*Then $\mathbf{z}_A$ is identifiable up to an element-wise invertible transformation and a permutation (Defn. 1), and $\mathbf{z}_B$ is identifiable up to a subspace-wise invertible transformation (Defn. 2).*

**Insights.** Theorem 2 demonstrates that, with sufficient diversity of the global structure, all class-dependent concepts can be identified up to element-wise indeterminacies. Notably, this result imposes no parametric constraints on the generative models or the nature of concepts, allowing for concept learning in a fully nonparametric setting. It also provides key insights into understanding nonlinear latent variable models without requiring additional prior knowledge.

**Discussion on Assumptions.** Assumption 1, referred to as *Structural Diversity*, ensures sufficient diversity across different classes of observations for the nonparametric identifiability of all class-dependent concepts. Without any parametric assumptions such as concept types, functional relations, or specific generative models, the only available information is the natural connective structure between classes and concepts. As previously discussed, if there is no diversity between classes, it becomes impossible to identify individual concepts without additional knowledge. Therefore, the *Structural Diversity* condition is essential for providing correctness guarantees for all concepts without relying on specific parametric assumptions or additional knowledge. Intuitively, it suggests that for each class-dependent concept $\mathbf{z}_i$, there exists a set of classes such that $\mathbf{z}_i$ is unique to one of these classes. For instance:

**Example 3.** *Consider $i = 1$ ($\mathbf{z}_1$ in Fig. 3). There exists a set of class indices $J = \{1, 3\}$ s.t. $M_{1,1} \neq 0$ and $M_{1,3} = 0$. Meanwhile, $M_{i,J\setminus\{j\}} = M_{1,3}$ is the only row with all zero entries in $M_{:,J\setminus\{j\}} = M_{:,3}$. Thus, the structural diversity holds for concept $\mathbf{z}_1$.*

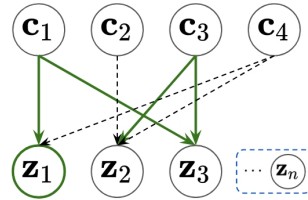

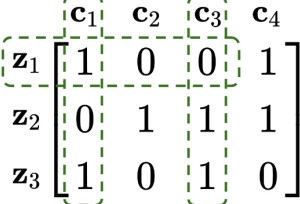

Figure 3: The *Structural Diversity* assumption, where the matrix represents $M$. Green lines indicate variables relevant to the discussion, while variables within the blue dotted square represent the class-independent variables $\mathbf{z}_B$.

Intuitively, the structural difference in the example above implies that $\mathbf{z}_1$ can be distinguished by considering these class indices. Simultaneously, we have sufficient information for all the remaining concepts, as the submatrix $M_{:,J\setminus 1}$ encompasses the other concepts. Consequently, it is possible to uniquely identify $\mathbf{z}_1$ among all the class-dependent hidden concepts. Coupled with this sufficient diversity for other concepts, we have the *Structural Diversity* assumption for the nonparametric identifiability of all class-dependent hidden concepts. In general, the proposed assumption necessitate the existence of diversity across classes in a structural way. Different

from various assumptions encouraging the sparsity of the structure in the literature (Rhodes & Lee, 2021; Moran et al., 2021; Zheng et al., 2022; Zheng & Zhang, 2023), our assumption only ensures necessary variability on the dependency structure and could also hold true with relatively dense connections. At the same time, we permit arbitrary structures between the class-dependent hidden concepts and the observed variables, while previous work has to assume a sparse structure on the generating process between latent and observed variables. This flexibility accommodates a general generative process, thereby distinguishing our assumptions from others. Additionally, another line of work on latent variable models requires $2n_A + 1$ distinct domains or classes to achieve latent variable identifiability (e.g., (Hyvärinen & Morioka, 2017; Khemakhem et al., 2020a; Kong et al., 2022; Hyvärinen et al., 2024)), a condition we do not impose.

Of course, since we aim for the general nonparametric identifiability for all class-dependent concepts, there are scenarios where it is impossible to fully recover every hidden concept, even with the help of the *Structural Diversity* condition. For instance, consider a scenario where all classes correspond to the same set of concepts, such as different breeds of dogs all sharing the concepts of "barks," "furry," and "four-legged." In this case, an infant or a machine without any prior knowledge would find it impossible to distinguish between the breeds based solely on these observational data. The lack of unique, distinguishing features for each breed means that the *Structural Diversity* condition cannot be satisfied, making it impossible to identify each breed's unique concepts purely from observation. This example highlights the limitations of the *Structural Diversity* condition in cases where inherent diversity across classes is absent. That being said, while the condition encourages diversity and can hold true in dense structures, it will fail if all concepts and classes are fully connected. In such a scenario, the lack of diversity between different classes makes it impossible to distinguish them without any extra information. In these instances, previous assumptions in provable concept learning—such as no occlusions between concepts (disjoint Jacobians), linear concept representations, and additive generating functions—can provide the additional information about the hidden process to ensure the identifiability of those concepts (Brady et al., 2023; Lachapelle et al., 2023; Wiedemer et al., 2024). Given this perspective, our assumption does *not* supersede the previous ones; rather, it offers a new direction that can be helpful for learning hidden concepts with minimal prior knowledge about the system.

The other assumption introduced in Thm. 2 requires distributional variability across different classes. Specifically, it necessitates the existence of at least two classes with differing conditional distributions. As discussed and empirically verified in Kong et al. (2022), the likelihood of *all* classes having identical probability measures is exceedingly slim. Importantly, these two classes may vary across different $A_{\mathbf{z}}$. Therefore, this assumption is highly likely to be satisfied in real-world scenarios, as it is virtually impossible for the measures corresponding to *all* classes (e.g., all kinds of animals in a zoo) to be almost identical. A concrete example is as follows:

**Example 4.** *Consider* $\mathbf{c}$ *as a 2-dimensional vector with* $c^{(k)} = [1,0]$ *and* $c^{(v)} = [0,1]$. *Let* $\mathcal{Z} = \mathbb{R}^2$, *and* $A_{\mathbf{z}} = \{(z_1, z_2) \in \mathbb{R}^2 : 0 \le z_1 \le 1, 0 \le z_2 \le 1\}$. *The conditional densities are* $p(\mathbf{z} \mid \mathbf{c} = [1,0]) = \frac{1}{2\pi} e^{-\frac{(z_1-1)^2+(z_2-0)^2}{2}}$ *and* $p(\mathbf{z} \mid \mathbf{c} = [0,1]) = \frac{1}{2\pi} e^{-\frac{(z_1-0)^2+(z_2-1)^2}{2}}$. *Evaluating the integrals over* $A_{\mathbf{z}}$, *we have*

$$\int_0^1 \int_0^1 \frac{1}{2\pi} e^{-\frac{(z_1-1)^2+(z_2-0)^2}{2}} dz_1 dz_2 \neq \int_0^1 \int_0^1 \frac{1}{2\pi} e^{-\frac{(z_1-0)^2+(z_2-1)^2}{2}} dz_1 dz_2.$$

*Note that* $(k,v)$ *can even be different for different* $A_{\mathbf{z}}$, *which further weakens the assumption.*

**Implications.** Extending the results on a subset of concepts (Thm. 1 and Prop. 1), Thm. 2 provides correctness guarantees for learning all class-dependent hidden concepts. Unlike previous work that focuses on specific parametric constraints such as disjointness, linearity, and additivity, the proposed global guarantees mainly rely on the *Structural Diversity* between classes and concepts, and thus can be applied on general scenarios given sufficient diversity. As discussed before, this aligns with the fundamental cognitive process of learning by comparison and ensures provably uncovering the latent world in a nonparametric manner. Despite being one of the essential pieces on learning the hidden concepts, our proposed theory also sheds light on understanding the latent variable models without additional knowledge, since the formulation is just based on the basic generating process between latent and observed variables. As a result, part of the proposed results might also be of independent interest to other fields such as disentanglement (Hyvärinen et al., 2024), causal representation

learning (Schölkopf et al., 2021), object-centric learning (Mansouri et al., 2024), compositional generalization (Du & Kaelbling, 2024), and causal structure learning (Spirtes et al., 2000).

**Class-independent concepts.** In Thm. 2, we have established the nonparametric identifiability of all class-dependent concepts. Similar to how infants learn about different objects by remembering their unique features, learning all concepts that do not always remain invariant might be sufficient for exploring the new world. However, we may still be interested in how to provably uncover the remaining class-independent concepts, even though they may not stand out in the cognitive process due to their invariance. Therefore, we provide the following result, with its proof in Appx. B.5, which identifies all concepts, whether class-dependent or class-independent, in a nonparametric manner.

**Proposition 2.** *Let the observed data be a sufficiently large sample generated by a model defined in Sec. 2. In addition to assumptions in Thm. 2, further suppose that, for all $\mathbf{z}_i \in \mathbf{z}_B$, there exists $\mathcal{C}_i$ s.t. $\bigcap_{k \in \mathcal{C}_i} \mathrm{supp}(D_{\mathbf{z}_i} f)_{i, n_A + 1:} = \{i\}$. Meanwhile, for each $i \in \{n_A + 1, \ldots, n\}$, there exist $\{\mathbf{z}^{(\ell)}\}_{\ell=1}^{|\mathcal{F}_{i, n_A + 1:}|}$ and a matrix $\mathrm{T}_f \in \mathcal{T}_f$ s.t. $\mathrm{span}\{D_{\mathbf{z}} f(\mathbf{z}^{(\ell)})_{i, n_A + 1:}\}_{\ell=1}^{|\mathcal{F}_{i, n_A + 1:}|} = \mathbb{R}^{n_B}_{\mathcal{F}_{i, n_A + 1:}}$ and $\left[D_{\mathbf{z}} f(\mathbf{z}^{(\ell)}) \mathrm{T}_f\right]_{i, n_A + 1:} \in \mathbb{R}^{n_B}_{\hat{\mathcal{F}}_{i, n_A + 1:}}$. Then $\mathbf{z}$ is identifiable up to an element-wise invertible transformation and a permutation (Defn. 1).*

To avoid introducing parametric assumptions, we still mainly rely on conditions on the connective structure. Since classes $\mathbf{c}$ are not connected to those class-independent concepts $\mathbf{z}_B$, the proposed structural condition on $M$ does not help identify $\mathbf{z}_B$. Thus, we leverage the structural condition between these concepts and the observed variables, as proposed in (Zheng et al., 2022). For brevity, let $\mathcal{F}$ and $\hat{\mathcal{F}}$ denote the support of the Jacobian $D_{\mathbf{z}} f$ and $D_{\hat{\mathbf{z}}} \hat{f}$, respectively. Additionally, $\mathcal{T}_f$ refers to a set of matrices with the same support of $\mathbf{T}_f$ in $D_{\hat{\mathbf{z}}} \hat{f} = D_{\mathbf{z}} f \mathbf{T}_f$, where $\mathbf{T}_f$ is a matrix-valued function. Generally, the condition on the structure $\mathrm{supp}(D_{\mathbf{z}_i} f)$ encourages sparsity in the Jacobian of the generating function $f$. As verified empirically in previous work (Zheng & Zhang, 2023), this condition is likely to hold in our setting where the number of observed variables $\mathbf{x}$ exceeds the number of class-independent concepts $\mathbf{z}_B$. Consequently, if needed, we can provide nonparametric guarantees under appropriate structural conditions for all types of concepts in general settings.

### 3.3 LEARNING STRUCTURE BETWEEN CLASSES AND CONCEPTS

Furthermore, we show that the hidden structure $M$, which encodes the dependency relations between classes and concepts, can also be identified based on multiple classes of observations (Prop. 3). This process parallels human learning, where distinguishing between classes involves recovering underlying structures, such as aligning concepts with their corresponding classes. Though identifying hidden structures in complex systems from observational data has remained an open problem for decades (Spirtes et al., 2000), our findings offer potential insights into addressing this longstanding challenge. The proof is included in Appx. B.4.

**Proposition 3.** *Let the observed data be a sufficiently large sample generated by a model defined in Sec. 2. Suppose all assumptions in Thm. 1 hold, except Assump. 1. Then the ground-truth structure $M$ is identifiable up to a row permutation.*

**Discussion on Assumptions.** All assumptions have been discussed in the previous sections. Compared to the previous theories on the identifiability of latent concepts, the recovery of the hidden connective structure does not necessitate the structural diversity assumption (Assump. 1). This allows us to uncover the structure in even more general scenarios, if the identification of latent concepts might not be of particular interest.

**Implications.** Proposition 3 indicates that, the recovered hidden structure between classes and concepts is an isomorphism of the ground-truth structure. Intuitively, this helps the machine understand which concepts correspond to a given class of observations. While this process may seem straightforward to us, it can be challenging for infants or machines without prior experience, as it aligns with an essential step of learning through comparison. For instance, consider an infant presented with a set of objects like a cat, a dog, and a bird (the classes) and a set of concepts like "furry," "barks," and "flies." Without proper knowledge, the infant might incorrectly assign "barks" to the cat or "flies" to the dog, lacking the experience to accurately match these concepts with the correct classes. The concept of "furry" might also be mistakenly assigned to the bird, despite its inapplicability. Therefore, to distinguish different classes by their concepts and learn unique concepts through

comparison, the machine must first recover the underlying connective structure. This is essential for provably learning from multiple classes of observations.

Furthermore, if we consider the class variables $\mathbf{c}$ as exogenous to the system and the underlying concept variables $\mathbf{z}$ as general hidden variables, the dependency structure between exogenous noises and hidden variables encodes most of the structural information in the system, even if dependencies exist among hidden variables (e.g., a hidden directed acyclic graph (DAG)). In structure learning, similar strategies have been applied to recover the DAG among hidden variables by first recovering the structure of how exogenous noises influence the system in both linear (Shimizu et al., 2006) and nonlinear (Reizinger et al., 2022) cases—the DAG constraint ensures the correspondence between the Jacobian of the mixing function and the adjacency matrix. It is worth noting that identifying the hidden structure in a general nonlinear system from purely observational data (i.e., without interventions) is a challenging problem that has been open for decades (Spirtes et al., 2000). Although this is not the focus of our work, the insights provided here may be of independent interest to researchers in related fields exploring this longstanding challenge.

## 4 EXPERIMENTS

In order to show the recovery of hidden concepts based on the proposed nonparametric identifiability theory, we conduct experiments on both synthetic and real-world datasets. It is noteworthy that an extensive body of research has empirically verified the ability to learn hidden concepts from various data modalities (Bau et al., 2017; Radford et al., 2017; Alvarez Melis & Jaakkola, 2018; Kim et al., 2018; Zhou et al., 2018; Yeh et al., 2020; Koh et al., 2020; Bai et al., 2022; Achtibat et al., 2022; Crabbé & van der Schaar, 2022; Liu et al., 2023). Furthermore, the application range of concept learning is expanding significantly with recent advancements in foundation models (Park et al., 2023; Rajendran et al., 2024; Jiang et al., 2024). Our results complement previous empirical findings by verifying the proposed theory, and we refer to the extensive previous research outlined above for more applications of concept learning across various scenarios.

**Setup.** In the considered setting, different samples may correspond to different classes selected by a mask. We structure the dataset as $\{(\mathbf{x}^{(i)}, \mathbf{c}^{(i)})\}_{i=1}^{N}$, where $N$ denotes the sample size, and $\mathbf{c}^{(i)}$ is a multi-hot vector representing the classes for the data point $\mathbf{x}^{(i)}$. A mask $\mathcal{M}_{i,:} \odot \mathbf{c}^{(i)}$ is applied to account for the specific class for each sample. We employ a regularized maximum-likelihood method during estimation, following the standard approach in (Sorrenson et al., 2020). The objective function is defined as $\mathcal{L}(\theta) = \mathbb{E}_{(\mathbf{x},\mathbf{c})}[\log p_{\hat{f}^{-1}}(\mathbf{x} \mid \mathcal{M}_{i,:} \odot \mathbf{c}) - \lambda \mathbf{R}]$, where $\lambda$ is the regularization parameter, and $\mathbf{R}$ represents the $\ell_1$ norm applied to $\hat{\mathcal{M}}$ and, if estimating class-independent concepts, also to $\hat{\mathcal{F}}$. Following previous work, we use Mean Correlation Coefficient (MCC) to measure the alignment between the ground-truth and the recovered latent concepts. The results are from 10 random trials. Additional details and results are provided in Appx. C.

**Synthetic datasets.** We conduct experiments on various synthetic datasets to verify the proposed identifiability theory. Specifically, we focus on two settings: learning all class-dependent concepts (Fig. 4) and learning all concepts, including class-independent ones, under appropriate conditions (Fig. 5). For *Ours*, the observations are generated according to the assumptions required for the theory; while for *Base*, no structural conditions on either $\mathcal{M}$ or $\mathcal{F}$ have been imposed. The details are included in Appx. C.1. Moreover, to measure the element-wise identifiability, we use the standard Mean Correlation Coefficient (MCC) between the ground-truth and estimated hidden concepts. The results (Fig. 4 and Fig. 5) demonstrate that our models achieve higher MCCs compared to the base model in both settings. This suggests that it is possible to identify hidden concepts from purely observational data without making assumptions about the concept type, functional relationships, or parametric generative models. Meanwhile, our models also provide lower variances across different runs, which further verifies our theoretical findings. As suggested by these results, hidden concepts can be identified up to an element-wise transformation and a permutation under our conditions, while the base model fails to disentangle and recover most concepts from data, further suggesting the necessity of the proposed conditions.

**Real-world datasets.** To assess the applicability of our proposed structural condition in real-world contexts, we performed experiments using the Fashion-MNIST (Xiao et al., 2017), EMNIST (Cohen et al., 2017), AnimalFace (Si & Zhu, 2011), and Flower102 (Nilsback & Zisserman, 2008) datasets. We highlight the identified concepts with the largest standard deviations (SDs) for Fashion-MNIST

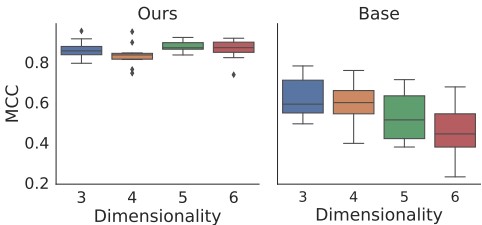

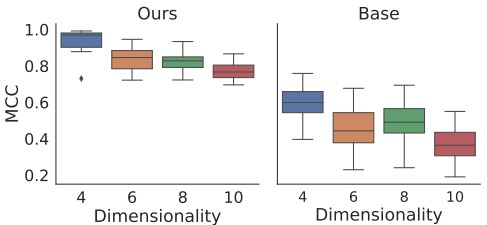

Figure 4: Identification of class-dependent concepts w.r.t. different number of concepts.

Figure 5: Identification of all concepts w.r.t. different number of concepts.

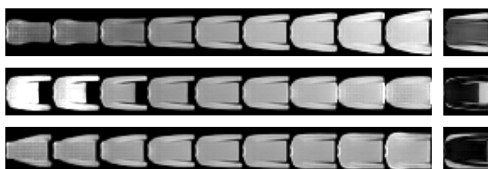

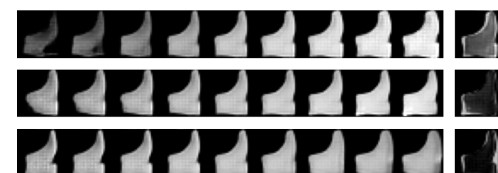

Figure 6: Results on Fashion-MNIST. The rows correspond to different concepts of a pullover: "sleeve length," "torso length," and "shoulder width," respectively.

Figure 7: Results on Fashion-MNIST. The rows correspond to different concepts of an ankle boot: "heel height," "ankle width," and "toe box width," respectively.

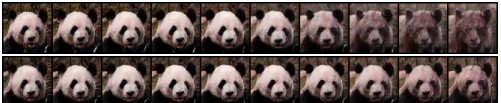

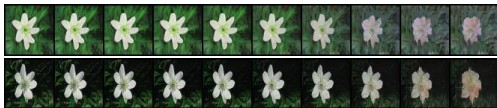

Figure 8: Results on AnimalFace. The rows correspond to different concepts of a panda: "Ursid" and "Monochrome," respectively.

Figure 9: Each row corresponds to the same concept ("Blooming") consistently identified from different environments in Flower102.

(Figs. 6 and 7), EMNIST (Fig. 10 in Appx. C.2), and AnimalFace (Fig. 8). Each row in the figures shows reconstructed images with the corresponding concept value varying to illustrate its effect. Additionally, the rightmost column features a heat map depicting the absolute pixel differences to visualize the influence. Clearly, the semantics of the identified concepts align with our understanding of the corresponding classes. For Flower102, we test the robustness of the recovered concept by comparing the same concept across different angles and environments. As seen in Fig. 9, the concept can be consistently identified from the same class across various conditions, further supporting our theory. Therefore, these results indicate that hidden concepts can be identified from observational data alone without the need to specify the generative model, underscoring the practical viability.

## 5 CONCLUSION

Drawing inspiration from the fundamental cognitive mechanism of learning through comparison, we establish a set of theoretical guarantees for learning concepts in general nonparametric settings. We provide a theoretical framework that potentially explains the impressive empirical successes in many previous works. Specifically, we prove that hidden concepts can be identified up to trivial indeterminacy from diverse classes of observations without any assumptions on the concept types, functional relations, or parametric generating models. Interestingly, even in scenarios where the structural conditions do not universally hold, we can still provide appropriate identifiability for a subset of concepts with sufficient diversity based on the mechanism of local comparison, thereby greatly broadening the applicability of the proposed theory. Furthermore, the connective structure between classes and concepts can also be recovered in a nonparametric manner. As a current limitation, future work involves exploiting the theory to a wider range of practical problems, such as compositional generalization, decision-making, and controllable generation.

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

# Appendices

## Table of Contents

## A  SUMMARY OF NOTATION

We summarize the key notations used throughout the paper to provide a quick reference for readers.

### VARIABLES AND FUNCTIONS

- $\mathbf{x} = (\mathbf{x}_1, \ldots, \mathbf{x}_m) \in \mathcal{X} \subseteq \mathbb{R}^m$ : Observed variables.

- $\mathbf{z} = (\mathbf{z}_A, \mathbf{z}_B) \in \mathcal{Z} \subseteq \mathbb{R}^n$, where $n = n_A + n_B$ : Latent concept variables.

- $\mathbf{z}_A \in \mathbb{R}^{n_A}$ : Class-dependent concepts influenced by the classes $\mathbf{c}$.

- $\mathbf{z}_B \in \mathbb{R}^{n_B}$ : Class-independent concepts, unaffected by $\mathbf{c}$.

- $\mathbf{c} = (\mathbf{c}_1, \ldots, \mathbf{c}_u)$ : Class variables represented as vectors, with $u$ classes.

- $f : \mathcal{Z} \to \mathcal{X}$ : Injective generative function mapping latent concepts to observations.

- $\mathbf{z}_A = g(\mathbf{c}, \theta, \epsilon)$ : Class-dependent concept function parameterized by $\mathbf{c}$, $\theta$ (factors), and $\epsilon$ (noise).

- $\theta$ : Additional influencing factors in the function $g$.

- $\epsilon$ : Noise term in the function $g$.

- $\hat{\mathbf{z}}$ : Estimated latent concepts.

- $\hat{f}$ : Estimated generative model.

### PROBABILITIES AND DENSITIES

- $p(\mathbf{z} \mid \mathbf{c}) = p(\mathbf{z}_A \mid \mathbf{c})p(\mathbf{z}_B)$ : Conditional density of latent concepts $\mathbf{z}$ given classes $\mathbf{c}$, assuming conditional independence.

- $p(\mathbf{z}_A \mid \mathbf{c}) = \prod_{i=1}^{n_A} p(\mathbf{z}_i \mid M_{i,:} \odot \mathbf{c})$ : Factorized density of class-dependent concepts $\mathbf{z}_A$.

- $\mathbb{E}[\cdot]$ : Expectation operator.

- $\mathbb{P}$ : Probability measure.

## INDICES AND SETS

- $A_i$ : Index set of concepts corresponding to class $\mathbf{c}_i$.

- $\mathbf{z}_{A_i}$ : Concepts associated with class $\mathbf{c}_i$.

- $\mathbf{z}_{A_i \setminus A_j}$ : Difference in concept sets between classes $\mathbf{c}_i$ and $\mathbf{c}_j$.

- $\mathcal{I} \subset \{1, \ldots, m\} \times \{1, \ldots, n\}$ : Set of indices for matrix elements.

- $\mathcal{I}_{i,:} = \{j \mid (i,j) \in \mathcal{I}\}$ : Indices corresponding to row $i$ in $\mathcal{I}$.

- $\mathcal{I}_{:,j} = \{i \mid (i,j) \in \mathcal{I}\}$ : Indices corresponding to column $j$ in $\mathcal{I}$.

- $\mathcal{S} \subset \{1, \ldots, n\}$ : Subset of indices.

- $\mathbb{R}^n_{\mathcal{S}} = \{s \in \mathbb{R}^n \mid s_i = 0 \text{ if } i \notin \mathcal{S}\}$ : Subspace of $\mathbb{R}^n$ where components not in $\mathcal{S}$ are zero.

## MATRICES AND OPERATIONS

- $S \in \mathbb{R}^{a \times b}$ : An arbitrary matrix with the shape $(a, b)$.

- $S_{i,:}, S_{:,j}$ : $i$-th row, $j$-th column of matrix $S$.

- $\operatorname{supp}(S) = \{(i,j) \mid S_{i,j} \neq 0\}$ : Support of matrix $S$.

- $\operatorname{supp}(\mathbf{S}(\Theta)) = \{(i,j) \mid \exists \theta \in \Theta, \mathbf{S}(\theta)_{i,j} \neq 0\}$ : Support of a matrix-valued function $\mathbf{S}(\Theta)$.

- $D_{\mathbf{c}}g$ : Partial derivative of $g$ with respect to class labels $\mathbf{c}$.

- $\mathcal{D} = \operatorname{supp}(D_{\mathbf{c}}g)$ : Support of the Jacobian of $g$ with respect to $\mathbf{c}$.

- $\mathbf{T}$ : Matrix-valued function representing a transformation between $D_{\mathbf{c}}g$ and $D_{\hat{\mathbf{c}}}\hat{g}$.

- $\mathcal{T}$ : Set of matrices sharing the same support as $\mathbf{T}$.

- $M \in \{0,1\}^{n_A \times u}$ : Binary structure matrix showing connections between classes and concepts.

- $\odot$ : Element-wise (Hadamard) product.

- $\operatorname{span}\{\cdot\}$ : Linear span of a set of vectors.

- $\operatorname{rank}(\cdot)$ : Rank of a matrix.

## DATA AND PARAMETERS

- $\{(\mathbf{x}^{(i)}, \mathbf{c}^{(i)})\}_{i=1}^N$ : Dataset of $N$ samples with observed variables and corresponding classes.

- $\mathcal{M}$ : Mask applied to classes in the dataset.

- $\lambda$ : Regularization parameter used in the estimation objective.

- $\mathbf{R}$ : Regularization term (e.g., $\ell_1$ norm applied to estimated supports).

- $\pi$ : Permutation function used to align estimated concepts.

- $\Theta$ : Parameter space.

## CONVENTIONS

- Bold lowercase letters (e.g., $\mathbf{x}$) denote vectors; uppercase letters (e.g., $S$, $M$) denote matrices.

- Calligraphic letters (e.g., $\mathcal{X}$, $\mathcal{Z}$) denote sets or spaces.

- Subscripts with colons denote slicing: $S_{i,:}$ represents the $i$-th row; $S_{:,j}$ represents the $j$-th column.

- Estimated quantities are denoted with hats (e.g., $\hat{\mathbf{z}}$ for estimated latent concepts).

# B PROOFS

## B.1 PROOF OF THEOREM 1

**Theorem 1.** *Let the observed data be a sufficiently large sample generated by a model defined in Sec. 2. Suppose for each $i \in \{1, \ldots, n_A\}$, there exist a set of points $\{(c, \theta)^{(\ell)}\}_{\ell=1}^{|\mathcal{D}_{:,i}|}$, a point $(c, \theta)^{(r)}$, and a matrix $\mathrm{T} \in \mathcal{T}$ such that the following conditions hold:*

> *i. The Jacobian spans its support space, i.e., $\mathrm{span}\{D_{\mathbf{c}}g((c,\theta)^{(\ell)})_{:,i}\}_{\ell=1}^{|\mathcal{D}_{:,i}|} = \mathbb{R}^{n_A}_{\mathcal{D}_{:,i}}$, and $\left[\mathrm{T}D_{\mathbf{c}}g((\mathbf{c},\theta)^{(\ell)})\right]_{:,i} \in \mathbb{R}^{n_A}_{\hat{\mathcal{D}}_{:,i}}$.*
>
> *ii. The Jacobian $D_{\mathbf{c}}g((\mathbf{c},\theta)^{(r)})$ is of full row rank.*

*Then for any two classes $\mathbf{c}_i$ and $\mathbf{c}_j$, there exists a permutation $\pi$ such that $\hat{\mathbf{z}}_{\pi(A_i \setminus A_j)}$, do not depend on the latent concepts $\mathbf{z}_{A_j}$ associated with class $\mathbf{c}_j$, and $\hat{\mathbf{z}}_{\pi(A_j \setminus A_i)}$ do not depend on the latent concepts $\mathbf{z}_{A_i}$ associated with class $\mathbf{c}_i$.*

*Proof.* Since both $D_{\mathbf{c}}g$ and $D_{\hat{\mathbf{c}}}\hat{g}$ are of full row rank, we have

$$D_{\hat{\mathbf{c}}}\hat{g} = \mathbf{T}D_{\mathbf{c}}g, \tag{4}$$

where $\mathbf{T}$ is an invertible matrix. According to the assumption, the span is nondegenerate in the sense that

$$\mathrm{span}\{D_{\mathbf{c}}g((c,\theta)^{(\ell)})_{:,j}\}_{\ell=1}^{|\mathcal{D}_{:,j}|} = \mathbb{R}^{n_A}_{\mathcal{D}_{:,j}}. \tag{5}$$

Then we can construct an one-hot vector $e_{i_0} \in \mathbb{R}^{n_A}_{\mathcal{D}_{:,j}}$ for any $i_0 \in \mathcal{D}_{:,j}$ as a linear combination of vectors $\{D_{\mathbf{c}}g((c,\theta)^{(\ell)})_{:,j}\}_{\ell=1}^{|\mathcal{D}_{:,j}|}$, i.e., $e_{i_0} = \sum_{\ell \in \mathcal{D}_{:,j}} \beta_\ell D_{\mathbf{c}}g((c,\theta)^{(\ell)})_{:,j}$, where $\beta_\ell$ denotes some coefficient. Note that we define $\mathcal{D}$ as the support of $D_{\mathbf{c}}g$. Additionally, we define $\mathcal{T}$ as a set of matrices that share the same support as $\mathbf{T}$ in the equation $D_{\hat{\mathbf{c}}}\hat{g} = \mathbf{T}D_{\mathbf{c}}g$, where $\mathbf{T}$ is a matrix-valued function and $\mathrm{T} \in \mathcal{T}$. Then we have

$$\mathrm{T}_{:,i_0} = \mathrm{T}e_{i_0} = \sum_{\ell \in \mathcal{D}_{:,j}} \beta_\ell \mathrm{T}D_{\mathbf{c}}g((c,\theta)^{(\ell)})_{:,j}. \tag{6}$$

According to the assumption, we have

$$\mathrm{T}D_{\mathbf{c}}g((c,\theta)^{(\ell)})_{:,j} \in \mathbb{R}^{n_A}_{\hat{\mathcal{D}}_{:,j}}. \tag{7}$$

Therefore, Eq. (6) implies $\mathrm{T}_{:,i_0} \in \mathbb{R}^{n_A}_{\hat{\mathcal{D}}_{:,j}}$, which is equivalent to

$$\forall i \in \mathcal{D}_{:,j}, \mathrm{T}_{:,i_0} \in \mathbb{R}^{n_A}_{\hat{\mathcal{D}}_{:,j}}. \tag{8}$$

This further indicates

$$\forall (i,j) \in \mathcal{D}, \mathcal{T}_{:,i} \times \{j\} \subset \hat{\mathcal{D}}. \tag{9}$$

Since $\mathbf{T}$ is invertible, we have

$$\det(\mathbf{T}) = \sum_{\sigma \in \mathcal{S}_{n_A}} \left(\mathrm{sgn}(\sigma) \prod_{j=1}^{n_A} \mathbf{T}_{\sigma(j),j}\right) \neq 0, \tag{10}$$

where $\mathcal{S}_{n_A}$ is a set of $n_A$-permutations. Then there must exist at least one non-zero term in the summation, which indicates that

$$\exists \sigma \in \mathcal{S}_{n_A}, \forall j \in \{1, \ldots, n_A\}, \mathrm{sgn}(\sigma) \prod_{j=1}^{n_A} \mathbf{T}_{\sigma(j),j} \neq 0. \tag{11}$$

Clearly, there cannot be any term in the product that equals zero, so we have

$$\exists \sigma \in \mathcal{S}_{n_A}, \forall j \in \{1, \ldots, n_A\}, \mathbf{T}_{\sigma(j),j} \neq 0. \tag{12}$$

Thus, it follows that

$$\forall i \in \{1, \ldots, n_A\}, \sigma(i) \in \mathcal{T}_{:,i}. \tag{13}$$

Then it yields

$$\forall (i,j) \in \mathcal{D}, (\sigma(i), j) \in \mathcal{T}_{:,i} \times \{j\}. \tag{14}$$

Because of Eq. (9), we have

$$\forall (i,j) \in \mathcal{D}, (\sigma(i), j) \in \hat{\mathcal{D}}. \tag{15}$$

Let us denote $\tilde{\pi}(\mathcal{D})$ as a row permutation of $\mathcal{D}$, where $\forall (i,j) \in \mathcal{D}$, there must be

$$(\sigma(i), j) \in \tilde{\pi}(\mathcal{D}) \tag{16}$$

and

$$|\tilde{\pi}(\mathcal{D})| = |\mathcal{D}|. \tag{17}$$

Furthermore, Eq. (15) indicates that

$$\tilde{\pi}(\mathcal{D}) \subset \hat{\mathcal{D}}, \tag{18}$$

We have the following relation based on the sparsity regularization:

$$|\hat{\mathcal{D}}| \leq |\mathcal{D}|. \tag{19}$$

Therefore, we have the following relation:

$$|\tilde{\pi}(\mathcal{D})| = |\mathcal{D}| \geq |\hat{\mathcal{D}}|. \tag{20}$$

Together with Eq. (18), it follows that

$$\hat{\mathcal{D}} = \tilde{\pi}(\mathcal{D}). \tag{21}$$

Let us denote the permutation indeterminacy in our goal as $\pi$ s.t.

$$\hat{\mathcal{D}} := \{(\pi(i), j) \mid (i,j) \in \mathcal{D}\}. \tag{22}$$

Given two classes $\mathbf{c}_i$ and $\mathbf{c}_j$, for any $\mathbf{z}_k \in \mathbf{z}_{A_i}$, we have

$$(k, i) \in \mathcal{D}. \tag{23}$$

Because of Eq. (9), this further implies

$$\mathcal{T}_{:,k} \times \{i\} \in \hat{\mathcal{D}}. \tag{24}$$

For any $\pi(v)$ where $\mathbf{z}_v \in \mathbf{z}_{A_j \setminus A_i}$, suppose we have

$$(\pi(v), k) \in \mathcal{T}, \tag{25}$$

which is equivalent to

$$\pi(v) \in \mathcal{T}_{:,k}. \tag{26}$$

Then according to Eq. (24), we have

$$(\pi(v), i) \in \mathcal{T}_{:,k} \times \{i\} \in \hat{\mathcal{D}}. \tag{27}$$

Based on Eq. (22), Eq. (27) is equivalent to

$$(v, i) \in \mathcal{D}, \tag{28}$$

which indicates a contradiction since $\mathbf{z}_v \in \mathbf{z}_{A_j \setminus A_i}$.

As a result, there must be $(\pi(v), k) \notin \mathcal{T}$. Similarly, for any $\mathbf{z}_u \in \mathbf{z}_{A_j}$, we can also show by contradiction that there must be $(\pi(u), j) \notin \mathcal{T}$. Therefore, for any two classes $\mathbf{c}_i$ and $\mathbf{c}_j$, there exists a permutation $\pi$ that the estimated latent concepts for the set difference, $\hat{\mathbf{z}}_{\pi(A_i \setminus A_j)}$, do not depend on the latent concepts $\mathbf{z}_{A_j}$ associated with class $\mathbf{c}_j$, and similarly, $\hat{\mathbf{z}}_{\pi(A_j \setminus A_i)}$ do not depend on of the latent concepts $\mathbf{z}_{A_i}$ associated with class $\mathbf{c}_i$. □

## B.2 PROOF OF PROPOSITION 1

**Proposition 1.** *Let the observed data be a sufficiently large sample generated by a model defined in Sec. 2. Suppose that the assumptions in Thm. 1 hold. Then, for a set of classes $\mathbf{c}_I$ and its corresponding concept sets $\mathbf{z}_{A_I}$ with a set of indices $I$, there exists a permutation $\pi$ that the unique part of a concept set for the class $\mathbf{c}_i$, i.e., $\hat{\mathbf{z}}_{\pi(A_i \setminus A_{I \setminus i})}$, does not depend on the latent concepts associated with other classes, i.e., $\mathbf{z}_{A_{I \setminus i}}$.*

*Proof.* Because all assumptions in Theorem 1 hold, according to the proof of it, we know that, for a row permutation of $\mathcal{D}$, i.e., $\tilde{\pi}(\mathcal{D})$ where

$$\tilde{\pi}(\mathcal{D}) := \{(\sigma(i), j) | (i, j) \in \mathcal{D}\}. \tag{29}$$

There must be a relationship that

$$\hat{\mathcal{D}} = \tilde{\pi}(\mathcal{D}). \tag{30}$$

Then we want to show that, there exists a permutation $\pi$ that the unique part of a concept set for the class $\mathbf{c}_i$, i.e., $\hat{\mathbf{z}}_{\pi(A_i \setminus A_{I \setminus i})}$, does not depend on the latent concepts associated with other classes, i.e., $\mathbf{z}_{A_{I \setminus i}}$. For any $z_k \in \mathbf{z}_{A_{I \setminus i}}$ and its corresponding class $c_q \in c_I$ and $q \neq i$, we have

$$(k, q) \in \mathcal{D}. \tag{31}$$

According to the proof of Theorem 1, we have

$$\mathrm{T} D_{\mathbf{c}} g((c, \theta)^{(\ell)})_{:,j} \in \mathbb{R}^{n_A}_{\hat{\mathcal{D}}_{:,j}}. \tag{32}$$

Therefore, Eq. (31) further indicates that

$$\mathcal{T}_{:,k} \times \{q\} \in \hat{\mathcal{D}}. \tag{33}$$

Define the permutation $\pi$ as

$$\hat{\mathcal{D}} := \{(\pi(i), j) \mid (i, j) \in \mathcal{D}\}. \tag{34}$$

Then we consider any $\pi(v)$ where we have

$$\mathbf{z}_v \in \mathbf{z}_{A_i \setminus A_{I \setminus i}}. \tag{35}$$

Suppose we have

$$(\pi(v), k) \in \mathcal{T}. \tag{36}$$

This also implies that

$$\pi(v) \in \mathcal{T}_{:,k}. \tag{37}$$

Based on Eq. (33), we further have

$$(\pi(v), q) \in \mathcal{T}_{:,k} \times \{q\} \in \hat{\mathcal{D}}. \tag{38}$$

According to the definition of $\hat{\mathcal{D}}$, this is equivalent to

$$(v, q) \in \mathcal{D}, \tag{39}$$

Because $\mathbf{z}_v \in \mathbf{z}_{A_i \setminus A_{I \setminus i}}$, the above equation indicates that there must be $c_q = c_i$. which is a contradiction since $q \neq i$. Therefore, we have

$$(\pi(v), k) \notin \mathcal{T}. \tag{40}$$

This implies that there exists a permutation $\pi$ that the unique part of a concept set for the class $\mathbf{c}_i$, i.e., $\hat{\mathbf{z}}_{\pi(A_i \setminus A_{I \setminus i})}$, does not depend on the latent concepts associated with other classes, i.e., $\mathbf{z}_{A_{I \setminus i}}$. $\square$

## B.3 PROOF OF THEOREM 2

**Theorem 2.** *Let the observed data be a sufficiently large sample generated by a model defined in Sec. 2. In addition to the assumptions in Thm. 1 and Assump. 1, suppose for any set $A_{\mathbf{z}} \subseteq \mathcal{Z}$ with non-zero probability measure and cannot be expressed as $B_{\mathbf{z}_B} \times \mathbf{z}_A$ for any $B_{\mathbf{z}_B} \subset \mathcal{Z}_B$, there exist two values of $\mathbf{c}$, i.e., $c^{(k)}$ and $c^{(v)}$ (which may vary across different $A_{\mathbf{z}}$), that*

$$\int_{\mathbf{z} \in A_{\mathbf{z}}} p(\mathbf{z} \mid c^{(k)}) d\mathbf{z} \neq \int_{\mathbf{z} \in A_{\mathbf{z}}} p(\mathbf{z} \mid c^{(v)}) d\mathbf{z}.$$

*Then $\mathbf{z}_A$ is identifiable up to an element-wise invertible transformation and a permutation (Defn. 1), and $\mathbf{z}_B$ is identifiable up to a subspace-wise invertible transformation (Defn. 2).*

*Proof.* Consider the transformation $h : \mathbf{z} \to \hat{\mathbf{z}}$ between true concepts $\mathbf{z}$ and estimated concepts $\hat{\mathbf{z}}$. Using the chain rule, the derivative of $\hat{g}$ with respect to $\hat{\mathbf{c}}$ can be expressed as:

$$D_{\hat{\mathbf{c}}}\hat{g} = D_{\mathbf{z}}h D_{\mathbf{c}}g. \tag{41}$$

The Jacobian of $h$ can be written as:

$$D_{\mathbf{z}}h = \left[\begin{array}{c|c} \frac{\partial \hat{\mathbf{z}}_A}{\partial \mathbf{z}_A} & \frac{\partial \hat{\mathbf{z}}_A}{\partial \mathbf{z}_B} \\ \hline \frac{\partial \hat{\mathbf{z}}_B}{\partial \mathbf{z}_A} & \frac{\partial \hat{\mathbf{z}}_B}{\partial \mathbf{z}_B} \end{array}\right]. \tag{42}$$

According to steps 1, 2, and 3 in the proof of Theorem 4.2 in Kong et al. (2022), the bottom-left block of $D_{\mathbf{z}}h$, i.e., $D_{\mathbf{z}}h_{n_A+1:,:n_A}$, consists of only zero entries. As a result, the Jacobian is equivalent to:

$$D_{\mathbf{z}}h = \left[\begin{array}{c|c} \frac{\partial \hat{\mathbf{z}}_A}{\partial \mathbf{z}_A} & \frac{\partial \hat{\mathbf{z}}_A}{\partial \mathbf{z}_B} \\ \hline \mathbf{0} & \frac{\partial \hat{\mathbf{z}}_B}{\partial \mathbf{z}_B} \end{array}\right]. \tag{43}$$

Since $h$ is invertible, the determinant of $D_{\mathbf{z}}h$ is non-zero. Together with the structure of the Jacobian matrix, we have

$$\det(D_{\mathbf{z}}h) = \det(\frac{\partial \hat{\mathbf{z}}_A}{\partial \mathbf{z}_A}) \det(\frac{\partial \hat{\mathbf{z}}_B}{\partial \mathbf{z}_B}), \tag{44}$$

which further implies

$$\det(\frac{\partial \hat{\mathbf{z}}_A}{\partial \mathbf{z}_A}) \neq 0, \tag{45}$$

$$\det(\frac{\partial \hat{\mathbf{z}}_B}{\partial \mathbf{z}_B}) \neq 0. \tag{46}$$

Since $\det(\frac{\partial \hat{\mathbf{z}}_B}{\partial \mathbf{z}_B}) \neq 0$ and $\frac{\partial \hat{\mathbf{z}}_B}{\partial \mathbf{z}_A} = 0$, it follows that $\hat{\mathbf{z}}_B$ depends solely on $\mathbf{z}_B$ and not on $\mathbf{z}_A$, i.e., there exists an invertible function $h_B : \mathbf{z}_B \to \hat{\mathbf{z}}_B$ s.t.,

$$\hat{\mathbf{z}}_B = h_B(\mathbf{z}_B). \tag{47}$$

Since $\hat{\mathbf{z}}_A$ is independent of $\hat{\mathbf{z}}_B$ and $\hat{\mathbf{z}}_B = h_B(\mathbf{z}_B)$, we further have $\hat{\mathbf{z}}_A$ is independent of $\mathbf{z}_B$, i.e.,

$$\frac{\partial \hat{\mathbf{z}}_A}{\partial \mathbf{z}_B} = 0. \tag{48}$$

Then the Jacobian can be represented as

$$D_{\mathbf{z}}h = \left[\begin{array}{c|c} \frac{\partial \hat{\mathbf{z}}_A}{\partial \mathbf{z}_A} & \mathbf{0} \\ \hline \mathbf{0} & \frac{\partial \hat{\mathbf{z}}_B}{\partial \mathbf{z}_B} \end{array}\right]. \tag{49}$$

Thus, $\hat{\mathbf{z}}_B$ is identifiable up to a subspace-wise invertible transformation, and we have

$$\begin{cases} \frac{\partial \hat{\mathbf{z}}_i}{\partial \mathbf{z}_j} = 0 & i \in \{1, \ldots, n_A\}, j \in \{n_A + 1, \ldots, n\}, \\ \frac{\partial \hat{\mathbf{z}}_k}{\partial \mathbf{z}_v} = 0 & k \in \{n_A + 1, \ldots, n\}, v \in \{1, \ldots, n_A\}. \end{cases} \tag{50}$$

This implies that

$$D_{\hat{\mathbf{c}}}\hat{g}_{:n_A,:} = D_{\mathbf{z}}h_{:n_A,:n_A} D_{\mathbf{c}}g_{:n_A,:}. \tag{51}$$

According to the assumption, we have

$$\text{span}\{D_{\mathbf{c}}g((c,\theta)^{(\ell)})_{:n_A,j}\}_{\ell=1}^{|\mathcal{D}_{:n_A,j}|} = \mathbb{R}^{n_A}_{\mathcal{D}_{:n_A,j}}. \tag{52}$$

Then we can construct an one-hot vector $e_{i_0} \in \mathbb{R}^{n_A}_{\mathcal{D}_{:n_A,j}}$ for any $i_0 \in \mathcal{D}_{:n_A,j}$ as a linear combination of vectors $\{D_{\mathbf{c}}g((c,\theta)^{(\ell)})_{:n_A,j}\}_{\ell=1}^{|\mathcal{D}_{:n_A,j}|}$, i.e., $e_{i_0} = \sum_{\ell \in \mathcal{D}_{:n_A,j}} \beta_\ell D_{\mathbf{c}}g((c,\theta)^{(\ell)})_{:n_A,j}$, where $\beta_\ell$ denotes some coefficient. Note that we define $\mathcal{T}$ as a set of matrices with the same support of $\mathbf{T}$ in $D_{\hat{\mathbf{c}}}\hat{g}_{:n_A,:} = \mathbf{T}D_{\mathbf{c}}g_{:n_A,:}$, where $\mathbf{T}$ is a matrix-valued function. Then we have

$$\mathbf{T}_{:,i_0} = \mathbf{T}e_{i_0} = \sum_{\ell \in \mathcal{D}_{:n_A,j}} \beta_\ell \mathbf{T}D_{\mathbf{c}}g((c,\theta)^{(\ell)})_{:n_A,j}. \tag{53}$$

According to the assumption, we have

$$\mathrm{T}D_{\mathbf{c}}g((c,\theta)^{(\ell)})_{:n_A,j} \in \mathbb{R}^{n_A}_{\hat{\mathcal{D}}_{:n_A,j}}. \tag{54}$$

Therefore, Eq. (53) implies $\mathrm{T}_{:,i_0} \in \mathbb{R}^{n_A}_{\hat{\mathcal{D}}_{:n_A,j}}$, which is equvalent to

$$\forall i \in \mathcal{D}_{:n_A,j}, \mathrm{T}_{:,i_0} \in \mathbb{R}^{n_A}_{\hat{\mathcal{D}}_{:n_A,j}}. \tag{55}$$

This further indicates

$$\forall (i,j) \in \mathcal{D}_{:n_A,:}, \mathcal{T}_{:,i} \times \{j\} \subset \hat{\mathcal{D}}_{:n_A,:}. \tag{56}$$

Since $\mathbf{T}$ is invertible, we have

$$\det(\mathbf{T}) = \sum_{\sigma \in \mathcal{S}_{n_A}} \left( \mathrm{sgn}(\sigma) \prod_{j=1}^{n_A} \mathbf{T}_{\sigma(j),j} \right) \neq 0, \tag{57}$$

where $\mathcal{S}_{n_A}$ is a set of $n_A$-permutations. Then there must exist at least one non-zero term in the summation, which indicates that

$$\exists \sigma \in \mathcal{S}_{n_A}, \forall j \in \{1, \ldots, n_A\}, \mathrm{sgn}(\sigma) \prod_{j=1}^{n_A} \mathbf{T}_{\sigma(j),j} \neq 0. \tag{58}$$

Clearly, there cannot be any term in the product that equals zero, so we have

$$\exists \sigma \in \mathcal{S}_{n_A}, \forall j \in \{1, \ldots, n_A\}, \mathbf{T}_{\sigma(j),j} \neq 0. \tag{59}$$

Thus, it follows that

$$\forall i \in \{1, \ldots, n_A\}, \sigma(i) \in \mathcal{T}_{:,i}. \tag{60}$$

Then it yields

$$\forall (i,j) \in \mathcal{D}_{:n_A,:}, (\sigma(i),j) \in \mathcal{T}_{:,i} \times \{j\}. \tag{61}$$

Because of Eq. (56), we have

$$\forall (i,j) \in \mathcal{D}_{:n_A,:}, (\sigma(i),j) \in \hat{\mathcal{D}}_{:n_A,:}. \tag{62}$$

Let us denote $\tilde{\pi}(\mathcal{D}_{:n_A,:})$ as a row permutation of $\mathcal{D}_{:n_A,:}$, where $\forall (i,j) \in \mathcal{D}_{:n_A,:}$, there must be

$$(\sigma(i),j) \in \tilde{\pi}(\mathcal{D}_{:n_A,:}), \tag{63}$$

and

$$|\tilde{\pi}(\mathcal{D}_{:n_A,:})| = |\mathcal{D}_{:n_A,:}|. \tag{64}$$

Eq. 62 indicates that

$$\tilde{\pi}(\mathcal{D}_{:n_A,:}) \subset \hat{\mathcal{D}}_{:n_A,:}. \tag{65}$$

According to the sparsity regularization, we have the following relation based on the sparsity regularization:

$$|\hat{\mathcal{D}}_{:n_A,:}| \leq |\mathcal{D}_{:n_A,:}|. \tag{66}$$

Therefore, we have

$$|\tilde{\pi}(\mathcal{D}_{:n_A,:})| = |\mathcal{D}_{:n_A,:}| \geq |\hat{\mathcal{D}}_{:n_A,:}|. \tag{67}$$

Together with Eq. (65), it follows that

$$\hat{\mathcal{D}}_{:n_A,:} = \tilde{\pi}(\mathcal{D}_{:n_A,:}). \tag{68}$$

Let us denote the permutation indeterminacy in our goal as $\pi$ s.t.

$$\hat{\mathcal{D}}_{:n_A,:} := \{(\pi(i),j) \mid (i,j) \in \mathcal{D}_{:n_A,:}\}. \tag{69}$$

For a latent concept $\mathbf{z}_i$, according to the structural diversity assumption (Assump. 1), there exists a set of column indices $J$, where $M_{i,J}$ only has one non-zero entry. Let us denote that non-zero entry as $M_{i,j}$. Since $M$ is a binary matrix with the support $\mathcal{D}_{:n_A,:}$, we have $(i,j) \in \mathcal{D}_{:n_A,:}$ and $(i,k) \notin \mathcal{D}_{:n_A,:}$ for any $k \in J \setminus j$.

Then, according to the assumption, for any other concept $\mathbf{z}_v$ where $v \neq i$, there must be a class $\mathbf{c}_q$ s.t. $q \in J \setminus j$ s.t.

$$(v,q) \in \mathcal{D}_{:n_A,:}. \tag{70}$$

Because of Eq. (56), it follows that

$$\mathcal{T}_{:,v} \times \{q\} \in \hat{\mathcal{D}}_{:n_A,:}. \tag{71}$$

For any $\pi(i)$, suppose we have

$$(\pi(i), v) \in \mathcal{T}, \tag{72}$$

which is equivalent to

$$\pi(i) \in \mathcal{T}_{:,v}. \tag{73}$$

Then according to Eq. (71), we have

$$(\pi(i), q) \in \mathcal{T}_{:,v} \times \{q\} \in \hat{\mathcal{D}}_{:n_A,:}. \tag{74}$$

Based on Eq. (69), Eq. (74) is equivalent to

$$(i, q) \in \mathcal{D}_{:n_A,:}. \tag{75}$$

This is a contradiction since $(i, q) \notin \mathcal{D}_{:n_A,:}$ for any $q \in J \setminus j$. Thus, for any $i \in \{1, \ldots, n_A\}$ and $k \in \{1, \ldots, n_A\} \setminus \{i\}$, there must be

$$(\pi(i), v) \notin \mathcal{T}. \tag{76}$$

Because $\mathcal{T}$ is invertible, all row must have at least one non-zero entry. Thus, Eq. (76) further implies

$$(\pi(i), i) \in \mathcal{T}. \tag{77}$$

Combining both Eqs. (76) and (77) for each $i \in \{1, \ldots, n_A\}$, the transformation between $\hat{\mathbf{z}}_A$ and $\mathbf{z}_A$ must be a composition of an element-wise invertible transformation and a permutation, which is our goal. $\qquad\square$

### B.4   PROOF OF PROPOSITION 3

**Proposition 3.** *Let the observed data be a sufficiently large sample generated by a model defined in Sec. 2. Suppose all assumptions in Thm. 1 hold, except Assump. 1. Then the ground-truth structure $M$ is identifiable up to a row permutation.*

*Proof.* Consider the transformation $h : \mathbf{z} \to \hat{\mathbf{z}}$ between true concepts $\mathbf{z}$ and estimated concepts $\hat{\mathbf{z}}$. Using the chain rule, the derivative of $\hat{g}$ with respect to $\hat{\mathbf{c}}$ can be expressed as:

$$D_{\hat{\mathbf{c}}}\hat{g} = D_{\mathbf{z}}h D_{\mathbf{c}}g. \tag{78}$$

The Jacobian of $h$ can be written as:

$$D_{\mathbf{z}}h = \left[\begin{array}{c|c} \frac{\partial \hat{\mathbf{z}}_A}{\partial \mathbf{z}_A} & \frac{\partial \hat{\mathbf{z}}_A}{\partial \mathbf{z}_B} \\ \hline \frac{\partial \hat{\mathbf{z}}_B}{\partial \mathbf{z}_A} & \frac{\partial \hat{\mathbf{z}}_B}{\partial \mathbf{z}_B} \end{array}\right]. \tag{79}$$

According to steps 1, 2, and 3 in the proof of Theorem 4.2 in Kong et al. (2022), the bottom-left block of $D_{\mathbf{z}}h$, i.e., $D_{\mathbf{z}}h_{n_A+1:,:n_A}$, consists of only zero entries. As a result, the Jacobian is equivalent to:

$$D_{\mathbf{z}}h = \left[\begin{array}{c|c} \frac{\partial \hat{\mathbf{z}}_A}{\partial \mathbf{z}_A} & \frac{\partial \hat{\mathbf{z}}_A}{\partial \mathbf{z}_B} \\ \hline \mathbf{0} & \frac{\partial \hat{\mathbf{z}}_B}{\partial \mathbf{z}_B} \end{array}\right]. \tag{80}$$

Since $h$ is invertible, the determinant of $D_{\mathbf{z}}h$ is non-zero. Together with the structure of the Jacobian matrix, we have

$$\det(D_{\mathbf{z}}h) = \det(\frac{\partial \hat{\mathbf{z}}_A}{\partial \mathbf{z}_A}) \det(\frac{\partial \hat{\mathbf{z}}_B}{\partial \mathbf{z}_B}), \tag{81}$$

which further implies

$$\det(\frac{\partial \hat{\mathbf{z}}_A}{\partial \mathbf{z}_A}) \neq 0, \tag{82}$$

$$\det(\frac{\partial \hat{\mathbf{z}}_B}{\partial \mathbf{z}_B}) \neq 0. \tag{83}$$

Since $\det(\frac{\partial \hat{\mathbf{z}}_B}{\partial \mathbf{z}_B}) \neq 0$ and $\frac{\partial \hat{\mathbf{z}}_B}{\partial \mathbf{z}_A} = 0$, it follows that $\hat{\mathbf{z}}_B$ depends solely on $\mathbf{z}_B$ and not on $\mathbf{z}_A$, i.e., there exists an invertible function $h_B : \mathbf{z}_B \rightarrow \hat{\mathbf{z}}_B$ s.t.,

$$\hat{\mathbf{z}}_B = h_B(\mathbf{z}_B). \tag{84}$$

Since $\hat{\mathbf{z}}_A$ is independent of $\hat{\mathbf{z}}_B$ and $\hat{\mathbf{z}}_B = h_B(\mathbf{z}_B)$, we further have $\hat{\mathbf{z}}_A$ is independent of $\mathbf{z}_B$, i.e.,

$$\frac{\partial \hat{\mathbf{z}}_A}{\partial \mathbf{z}_B} = 0. \tag{85}$$

Therefore, the Jacobian of $h$ is

$$D_{\mathbf{z}}h = \left[ \begin{array}{c|c} \frac{\partial \hat{\mathbf{z}}_A}{\partial \mathbf{z}_A} & \mathbf{0} \\ \hline \mathbf{0} & \frac{\partial \hat{\mathbf{z}}_B}{\partial \mathbf{z}_B} \end{array} \right]. \tag{86}$$

Note that we have

$$D_{\hat{\mathbf{c}}}\hat{g} = D_{\mathbf{z}}h D_{\mathbf{c}}g, \tag{87}$$

which is equivalent to

$$D_{\hat{\mathbf{c}}}\hat{g}_{:n_A,:} = (D_{\mathbf{z}}h D_{\mathbf{c}}g)_{:n_A,:} = D_{\mathbf{z}}h_{:n_A,:}D_{\mathbf{c}}g. \tag{88}$$

Because $\frac{\partial \hat{\mathbf{z}}_i}{\partial \mathbf{z}_k} = 0$ for $i \in \{1, \ldots, n_A\}$ and $k \in \{n_A + 1, \ldots, n\}$, the upper-right block of $D_{\mathbf{z}}h$, i.e., $D_{\mathbf{z}}h_{:n_A,n_A+1:}$, consists of only zero entries. It further indicates that

$$D_{\hat{\mathbf{c}}}\hat{g}_{:n_A,:} = D_{\mathbf{z}}h_{:n_A,:n_A}D_{\mathbf{c}}g_{:n_A,:}. \tag{89}$$

According to the assumption, we have

$$\mathrm{span}\{D_{\mathbf{c}}g((c,\theta)^{(\ell)})_{:n_A,j}\}_{\ell=1}^{|\mathcal{D}_{:n_A,j}|} = \mathbb{R}^{n_A}_{\mathcal{D}_{:n_A,j}}. \tag{90}$$

Then we can construct an one-hot vector $e_{i_0} \in \mathbb{R}^{n_A}_{\mathcal{D}_{:n_A,j}}$ for any $i_0 \in \mathcal{D}_{:n_A,j}$ as a linear combination of vectors $\{D_{\mathbf{c}}g((c,\theta)^{(\ell)})_{:n_A,j}\}_{\ell=1}^{|\mathcal{D}_{:n_A,j}|}$, i.e., $e_{i_0} = \sum_{\ell \in \mathcal{D}_{:n_A,j}} \beta_\ell D_{\mathbf{c}}g((c,\theta)^{(\ell)})_{:n_A,j}$, where $\beta_\ell$ denotes some coefficient. Then we have

$$\mathrm{T}_{:,i_0} = \mathrm{T}e_{i_0} = \sum_{\ell \in \mathcal{D}_{:n_A,j}} \beta_\ell \mathrm{T}D_{\mathbf{c}}g((c,\theta)^{(\ell)})_{:n_A,j}. \tag{91}$$

Note that we define $\mathcal{D}$ as the support of $D_{\mathbf{c}}g$. Additionally, we define $\mathcal{T}$ as a set of matrices that share the same support as $\mathbf{T}$ in the equation $D_{\hat{\mathbf{c}}}\hat{g}_{:n_A,:} = \mathbf{T}D_{\mathbf{c}}g_{:n_A,:}$, where $\mathbf{T}$ is a matrix-valued function and $\mathbf{T} \in \mathcal{T}$.

According to the assumption, we have

$$\mathrm{T}D_{\mathbf{c}}g((c,\theta)^{(\ell)})_{:n_A,j} \in \mathbb{R}^{n_A}_{\hat{\mathcal{D}}_{:n_A,j}}. \tag{92}$$

Therefore, Eq. (91) implies $\mathrm{T}_{:,i_0} \in \mathbb{R}^{n_A}_{\hat{\mathcal{D}}_{:n_A,j}}$, which is equivalent to

$$\forall i_0 \in \mathcal{D}_{:n_A,j}, \mathrm{T}_{:,i_0} \in \mathbb{R}^{n_A}_{\hat{\mathcal{D}}_{:n_A,j}}. \tag{93}$$

This further indicates

$$\forall (i,j) \in \mathcal{D}_{:n_A,:}, \mathcal{T}_{:,i} \times \{j\} \subset \hat{\mathcal{D}}_{:n_A,:}. \tag{94}$$

Since $\mathbf{T}$ is invertible, we have

$$\det(\mathbf{T}) = \sum_{\sigma \in \mathcal{S}_{n_A}} \left( \mathrm{sgn}(\sigma) \prod_{j=1}^{n_A} \mathbf{T}_{\sigma(j),j} \right) \neq 0, \tag{95}$$

where $\mathcal{S}_{n_A}$ is a set of $n_A$-permutations. Then there must exist at least one non-zero term in the summation, which indicates that

$$\exists \sigma \in \mathcal{S}_{n_A}, \forall j \in \{1, \ldots, n_A\}, \mathrm{sgn}(\sigma) \prod_{j=1}^{n_A} \mathbf{T}_{\sigma(j),j} \neq 0. \tag{96}$$

Clearly, there cannot be any term in the product that equals zero, so we have

$$\exists \sigma \in \mathcal{S}_{n_A}, \forall j \in \{1, \ldots, n_A\}, \mathbf{T}_{\sigma(j),j} \neq 0. \tag{97}$$

Thus, it follows that

$$\forall i \in \{1, \ldots, n_A\}, \sigma(i) \in \mathcal{T}_{:,i}. \tag{98}$$

Then it yields

$$\forall (i,j) \in \mathcal{D}_{:n_A,:}, (\sigma(i), j) \in \mathcal{T}_{:,i} \times \{j\}. \tag{99}$$

Because of Eq. (94), we have

$$\forall (i,j) \in \mathcal{D}_{:n_A,:}, (\sigma(i), j) \in \hat{\mathcal{D}}_{:n_A,:}. \tag{100}$$

Let us denote $\pi(\mathcal{D}_{:n_A,:})$ as a row permutation of $\mathcal{D}_{:n_A,:}$, where $\forall (i,j) \in \mathcal{D}_{:n_A,:}$, there must be

$$(\sigma(i), j) \in \pi(\mathcal{D}_{:n_A,:}). \tag{101}$$

And it also implies

$$|\pi(\mathcal{D}_{:n_A,:})| = |\mathcal{D}_{:n_A,:}|. \tag{102}$$

Furthermore, Eq. 100 indicates that

$$\pi(\mathcal{D}_{:n_A,:}) \subset \hat{\mathcal{D}}_{:n_A,:}, \tag{103}$$

We have the following relation based on the sparsity regularization:

$$|\hat{\mathcal{D}}_{:n_A,:}| \leq |\mathcal{D}_{:n_A,:}|. \tag{104}$$

Therefore, we have

$$|\pi(\mathcal{D}_{:n_A,:})| = |\mathcal{D}_{:n_A,:}| \geq |\hat{\mathcal{D}}_{:n_A,:}|. \tag{105}$$

Together with Eq. (103), it follows that

$$\hat{\mathcal{D}}_{:n_A,:} = \pi(\mathcal{D}_{:n_A,:}). \tag{106}$$

Thus, we have proved the identifiability of $\mathcal{D}_{:n_A,:}$ up to a permutation on the row indices. Since $M$ is a binary matrix with the support of $\mathcal{D}$, we have proved the connective structure between classes and concepts up to a row permutation. □

## B.5 PROOF OF PROPOSITION 2

**Proposition 2.** *Let the observed data be a sufficiently large sample generated by a model defined in Sec. 2. In addition to assumptions in Thm. 2, further suppose that, for all $\mathbf{z}_i \in \mathbf{z}_B$, there exists $\mathcal{C}_i$ s.t. $\bigcap_{k \in \mathcal{C}_i} \mathrm{supp}(D_{\mathbf{z}_i}f)_{i,n_A+1:} = \{i\}$. Meanwhile, for each $i \in \{n_A+1, \ldots, n\}$, there exist $\{\mathbf{z}^{(\ell)}\}_{\ell=1}^{|\mathcal{F}_{i,n_A+1:}|}$ and a matrix $\mathrm{T}_f \in \mathcal{T}_f$ s.t. $\mathrm{span}\{D_{\mathbf{z}}f(\mathbf{z}^{(\ell)})_{i,n_A+1:}\}_{\ell=1}^{|\mathcal{F}_{i,n_A+1:}|} = \mathbb{R}^{n_B}_{\mathcal{F}_{i,n_A+1:}}$ and $[D_{\mathbf{z}}f(\mathbf{z}^{(\ell)})\mathrm{T}_f]_{i,n_A+1:} \in \mathbb{R}^{n_B}_{\hat{\mathcal{F}}_{i,n_A+1:}}$. Then $\mathbf{z}$ is identifiable up to an element-wise invertible transformation and a permutation (Defn. 1).*

*Proof.* We denote the transformation between the true and estimated concepts as $h : \mathbf{z} \to \hat{\mathbf{z}}$. According to the proof in Theorem 2, the Jacobian $h$ is as follows:

$$D_{\mathbf{z}}h = \left[\begin{array}{c|c} \dfrac{\partial \hat{\mathbf{z}}_A}{\partial \mathbf{z}_A} & \mathbf{0} \\ \hline \mathbf{0} & \dfrac{\partial \hat{\mathbf{z}}_B}{\partial \mathbf{z}_B} \end{array}\right]. \tag{107}$$

Therefore, any variable in $\hat{\mathbf{z}}_A$ does not depend on any variable in $\mathbf{z}_B$, and any variable in $\hat{\mathbf{z}}_B$ does not depend on any variable in $\mathbf{z}_A$. At the same time, by using the chain rule on $h = \hat{f}^{-1} \circ f$, we have

$$D_{\hat{\mathbf{z}}}\hat{f} = D_{\mathbf{z}}f D_{\hat{\mathbf{z}}}h^{-1}, \tag{108}$$

which is equivalent to

$$D_{\hat{\mathbf{z}}}\hat{f}_{:,n_A+1:} = D_{\mathbf{z}}f D_{\hat{\mathbf{z}}}h^{-1}_{:,n_A+1:}. \tag{109}$$

Based on Eq. 107, this further indicates that

$$D_{\hat{\mathbf{z}}}\hat{f}_{:,n_A+1:} = D_{\mathbf{z}}f_{:,n_A+1:}D_{\hat{\mathbf{z}}}h^{-1}_{n_A+1:,n_A+1:}. \tag{110}$$

Then we have the following equation according to the assumption:

$$\text{span}\{D_{\mathbf{z}}f(\mathbf{z}^{(\ell)})_{i,n_A+1:}\}_{\ell=1}^{|\mathcal{F}_{i,n_A+1:}|} = \mathbb{R}^{n_B}_{\mathcal{F}_{i,n_A+1:}} \tag{111}$$

Then we can construct an one-hot vector $e_{j_0} \in \mathbb{R}^{n_B}_{\mathcal{F}_{i,n_A+1:}}$ for any $j_0 \in \mathcal{F}_{i,n_A+1:}$ as a linear combination of vectors $\{D_{\mathbf{z}}f(\mathbf{z}^{(\ell)})_{i,n_A+1:}\}_{\ell=1}^{|\mathcal{F}_{i,n_A+1:}|}$, i.e.,

$$e_{j_0} = \sum_{\ell \in \mathcal{F}_{i,n_A+1:}} \beta_\ell D_{\mathbf{z}}f(\mathbf{z}^{(\ell)})_{i,n_A+1:}, \tag{112}$$

where $\beta_\ell$ denotes some coefficient. Then we have

$$\mathrm{T}_{f_{j_0,n_A+1:}} = e_{j_0}\mathrm{T}_{f_{:,n_A+1:}} = \sum_{\ell \in \mathcal{D}_{:n_A,j}} \beta_\ell D_{\mathbf{z}}f(\mathbf{z}^{(\ell)})_{i,n_A+1:}\mathrm{T}_{f_{:,n_A+1:}} \in \mathbb{R}^{n_B}_{\hat{\mathcal{F}}_{i,n_A+1:}}. \tag{113}$$

This further implies that, for any $j \in \mathcal{F}_{i,n_A+1:}$, we always have $\mathrm{T}_{f_{j,:}} \in \mathbb{R}^{n_B}_{\hat{\mathcal{F}}_{i,n_A+1:}}$. Thus, we have the connection between support as follows:

$$(i,j) \in \mathcal{F}_{:,n_A+1:}, \{i\} \times \mathcal{T}_{f_{j,:}} \subset \hat{\mathcal{F}}_{:,n_A+1:}. \tag{114}$$

Then, because of the invertibility of $\mathbf{T}_f$, its determinant must not equal to zero, i.e.,

$$\sum_{\sigma \in \mathcal{S}_n} \left( \text{sgn}(\sigma) \prod_{i=1}^{n_B} \mathbf{T}_f(\mathbf{z}^{(\ell)})_{i,\sigma(i)} \right) \neq 0, \tag{115}$$

where $\mathcal{S}$ is the set of $n$-permutations. Therefore, there must be at least one term in the summation that does not equal to zero, i.e.,

$$\exists \sigma \in \mathcal{S}_n, \forall i \in \{1,\ldots,n_B\}, \text{sgn}(\sigma) \prod_{i=1}^{n_B} \mathbf{T}_f(\mathbf{z}^{(\ell)})_{i,\sigma(i)} \neq 0. \tag{116}$$

Because $\text{sgn}(\sigma) \neq 0$, every term in the production must not equal to zero, i.e.,

$$\exists \sigma \in \mathcal{S}_n, \forall i \in \{1,\ldots,n_B\}, \mathbf{T}_f(\mathbf{z}^{(\ell)})_{i,\sigma(i)} \neq 0. \tag{117}$$

This follows that

$$\forall j \in \{1,\ldots,n_B\}, \sigma(j) \in \mathcal{T}_{f_{j,n_A+1:}}. \tag{118}$$

Based on Eq. (114), Eq. (118) further implies that, for any $(i,j) \in \mathcal{F}_{:,n_A+1:}$, we have $(i,\sigma(j)) \in \hat{\mathcal{F}}_{:,n_A+1:}$. Let us denote $\sigma(\mathcal{F}) = \{(i,\sigma(j)) \mid (i,j) \in \mathcal{F}\}$, the above connection implies $\sigma(\mathcal{F}) \subset \hat{\mathcal{F}}$. Together with the sparsity regularization on the estimated Jacobian, we have

$$|\hat{\mathcal{F}}| \leq |\mathcal{F}| \tag{119}$$

Because of the definition of $\sigma(\mathcal{F})$, there must be

$$|\mathcal{F}| = |\sigma(\mathcal{F})|, \tag{120}$$

which follows that

$$|\sigma(\mathcal{F})| \geq |\hat{\mathcal{F}}|. \tag{121}$$

Together with the relation that $\sigma(\mathcal{F}) \subset \hat{\mathcal{F}}$, there must be

$$\hat{\mathcal{F}} = \sigma(\mathcal{F}). \tag{122}$$

Suppose $\mathbf{T}_{:,n_A+1:}$ is not a composition of a permutation matrix and a diagonal matrix, then

$$\exists j_1 \neq j_2, \mathcal{T}_{j_1,n_A+1:} \cap \mathcal{T}_{j_2,n_A+1:} \neq \emptyset. \tag{123}$$

Additionally, consider $j_3 \in \{1,\ldots,n_B\}$ for which

$$\sigma(j_3) \in \mathcal{T}_{j_1,n_A+1:} \cap \mathcal{T}_{j_2,n_A+1:}. \tag{124}$$

Since $j_1 \neq j_2$, we can assume $j_3 \neq j_1$ without loss of generality. Based on assumption, there exists $\mathcal{C}_{j_1} \ni j_1$ such that $\bigcap_{i \in \mathcal{C}_{j_1}} \mathcal{F}_{i,n_A+1:} = \{j_1\}$. Because

$$j_3 \notin \{j_1\} = \bigcap_{i \in \mathcal{C}_{j_1}} \mathcal{F}_{i,n_A+1:}, \tag{125}$$

there must exists $i_3 \in \mathcal{C}_{j_1}$ such that

$$j_3 \notin \mathcal{F}_{i_3,n_A+1:}. \tag{126}$$

Since $j_1 \in \mathcal{F}_{i_3,n_A+1:}$, it follows that $(i_3, j_1) \in \mathcal{F}_{:,n_A+1:}$. Therefore, according to Eq. (114), we have

$$\{i_3\} \times \mathcal{T}_{j_1,n_A+1:} \subset \hat{\mathcal{F}}_{:,n_A+1:}. \tag{127}$$

Notice that $\sigma(j_3) \in \mathcal{T}_{j_1,n_A+1:} \cap \mathcal{T}_{j_2,n_A+1:}$ implies

$$(i_3, \sigma(j_3)) \in \{i_3\} \times \mathcal{T}_{j_1,n_A+1:}. \tag{128}$$

Then by Eqs. (127) and (128), we have

$$(i_3, \sigma(j_3)) \in \hat{\mathcal{F}}_{:,n_A+1:}. \tag{129}$$

This further implies $(i_3, j_3) \in \mathcal{F}_{:,n_A+1:}$ by Eq. (122), which contradicts Eq. (126). Therefore, we have proven by contradiction that $\mathbf{T}_{:,n_A+1:}$ is a composition of a permutation matrix and a diagonal matrix, which means that the invariant part $\mathbf{z}_B$ is identifiable up to an element-wise invertible transformation and a permutation. Together with the element-wise identifiability for concepts in the changing part $\mathbf{z}_A$ given by Theorem 2, we have proved that all latent concepts $\mathbf{z} = (\mathbf{z}_A, \mathbf{z}_B)$ is identifiable up to an element-wise invertible transformation and a permutation. $\square$

## C   EXPERIMENTS

In this section, we provide more details regarding the experimental setup as well as additional experimental results to further support our theoretical findings.

### C.1   SUPPLEMENTARY EXPERIMENTAL SETUP

We generate the data following the process outlined in our theorems. For our model that identifies only class-dependent concepts (Fig. 4), the connective structure between classes and concepts is generated according to the *Structural Diversity* condition. For class-dependent concepts, we sample from two multivariate Gaussian distributions with zero means and variances drawn from a uniform distribution on $[0.5, 3]$, consistent with parameters used in previous work (Khemakhem et al., 2020b; Sorrenson et al., 2020). For our model that identifies all hidden concepts, including class-independent ones (Fig. 5), the connective structure between class-independent concepts and observed variables follows the structural condition in Prop. 2. These class-independent concepts are sampled from a single multivariate Gaussian distribution with zero means and variances drawn from a uniform distribution on $[0.5, 3]$. In the base model, we remove the structural constraints on both types of connective structures to verify the necessity of the proposed conditions. All other settings remain the same as ours.

In our model evaluation, we employ the Mean Correlation Coefficient (MCC) to measure the alignment between the ground-truth and the recovered latent concepts, which is standard in the literature (Hyvärinen & Morioka, 2016). To calculate MCC, we first compute the pairwise correlation coefficients between the true concepts and the recovered concepts after applying a component-wise transformation via regression. Following this, we solve an assignment to match each recovered concept to the corresponding ground-truth concept with the highest correlation.

We use Generative Flow (Kingma & Dhariwal, 2018) as the nonlinear generating function. For synthetic settings, the sample size is set as $10,000$. Experiments are conducted using the official implementation of GIN[2] (Sorrenson et al., 2020) with an additional $\ell_1$ regularization on the Jacobians and FrEIA[3] (Ardizzone et al., 2018-2022) for the flow-based generative function. The regularization parameters $\lambda$ is set according to a search in $\lambda \in \{0.01, 0.1, 1\}$, and we select $\lambda = 0.1$ according to the average MCCs of experiments conducted on synthetic datasets. Moreover, all experiments are conducted on 12 CPU cores with 16 GB RAM.

---

[2]https://github.com/VLL-HD/GIN
[3]https://github.com/vislearn/FrEIA

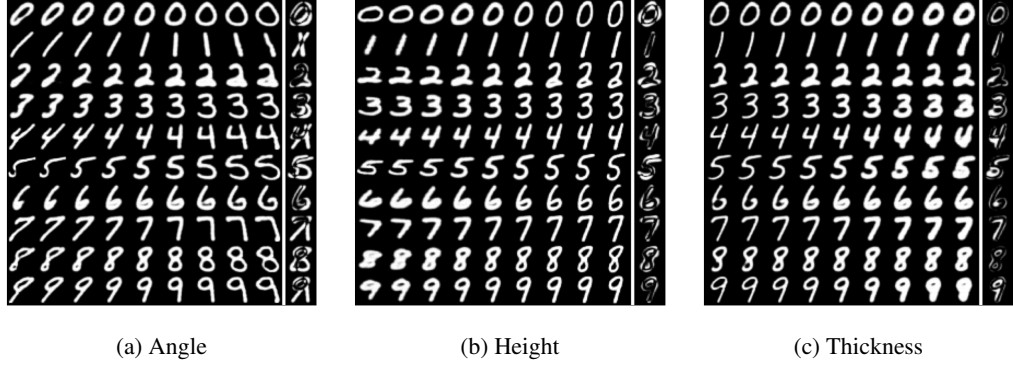

(a) Angle             (b) Height             (c) Thickness

Figure 10: Results for each digit class in the EMNIST dataset, showing the identified concepts with the top three standard deviations (SDs). Each subfigure represents a concept identified by our model, with values ranging from $-4$ to $+4$ SDs to demonstrate their impact. The rightmost column features a heat map of the absolute pixel differences between $-1$ and $+1$ SDs. These concepts can be interpreted as variations in angle, height, and thickness.

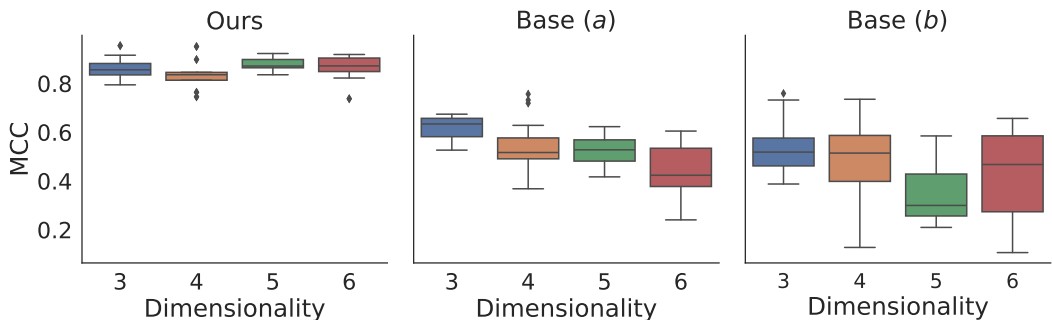

Figure 11: Identification of concepts w.r.t. different number of concepts and different settings.

## C.2 SUPPLEMENTARY EXPERIMENTAL RESULTS

As discussed in Sec. 4, we include the results on EMNIST dataset here in the appendix. The EMNIST dataset (Cohen et al., 2017) is an extension of the classical MNIST, which consists of a much larger set of handwritten digits derived from the NIST Special Database 19 (Grother & Hanaoka, 1995).

The results are shown in Fig. 10. Similar to the other datasets, we select the identified components with the top three standard deviations and vary the value of the identified components to visualize their potential semantics. According to the results, it is clear that the hidden concepts can be identified by only learning from diverse classes of observations. This further indicates that the proposed nonparametric identifiability, which is based on the basic cognitive mechanism of learning by comparison, has potential applicability in real-world scenarios.

**Partial violation of previous conditions.** We also conduct experiments to evaluate the identification under partial violations of previously established assumptions in the literature of latent variable models. Specifically, we generated datasets with the following conditions:

1. *Base (a)*: The structural sparsity assumption on the mixing structure between latent concepts and observed variables, as outlined in (Zheng et al., 2022; Zheng & Zhang, 2023), is partially violated for a subset of concepts, with the size randomly selected from all integers in the range 1 to $n/2$.

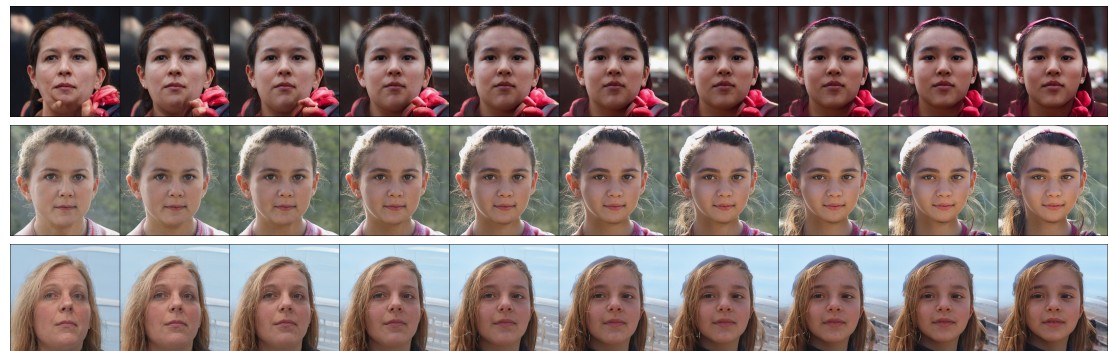

Figure 12: Multiple concepts (e.g., skin, eyes, face shape, etc.) corresponding to "Age" are entangled after estimation.

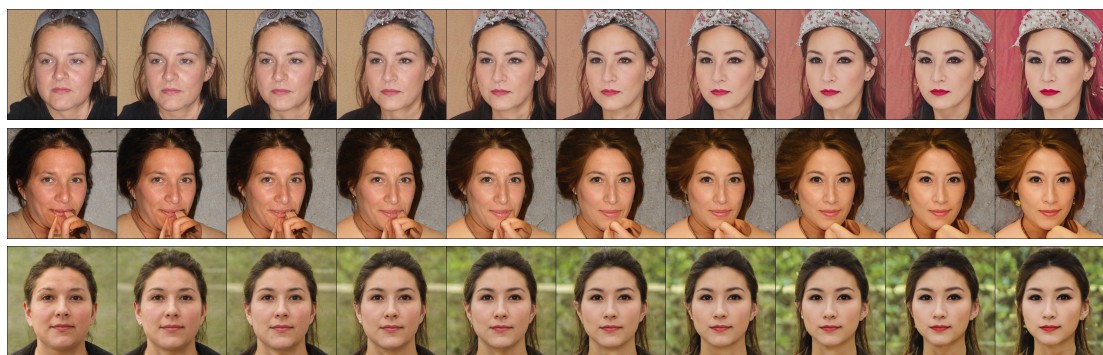

Figure 13: Multiple concepts (e.g., lipstick, eye shadow, powder, etc.) corresponding to "Makeup" are entangled after estimation.

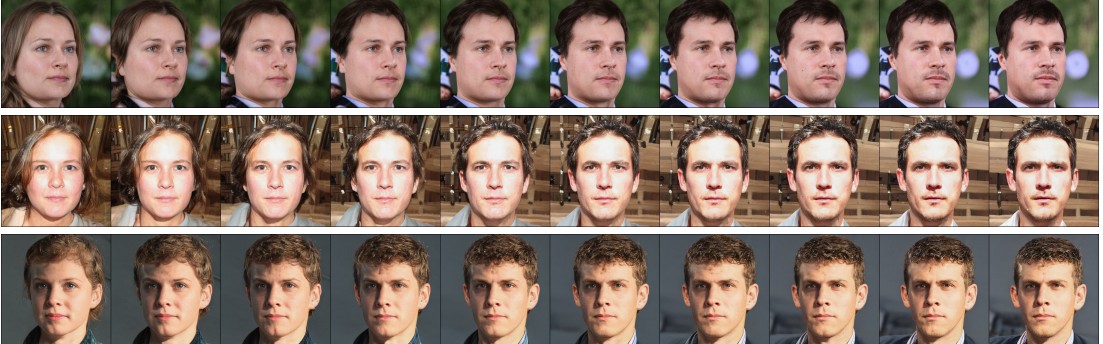

Figure 14: Multiple concepts (e.g., hairstyle, head shape, eye, etc.) corresponding to "Gender" are entangled after estimation.

2. *Base* ($b$): The $2n + 1$ domain requirement in (Kong et al., 2022) is partially violated. Instead, latent concepts are generated from $n + 1$ multivariate Gaussian distributions, each with zero mean and variances drawn from a uniform distribution over $[0.5, 3]$.

3. *Ours*: The data-generating process adheres to our proposed structural diversity condition. While there are no constraints on the mixing structure between latent concepts and observed variables, the structure between classes and concepts satisfies the required structural diversity.

The results, shown in Fig. 11, indicate that when assumptions from previous works are partially violated, the recovery of latent concepts becomes unreliable. This demonstrates the sensitivity of prior methods to these assumptions. All results are from 10 runs with different random seeds.

**Additional real-world experiments.** To explore scenarios where not all concepts can be identified component-wise, we conduct additional real-world experiments on a more complex scenario, i.e., the FFHQ dataset (Karras et al., 2019). The dataset contains $70,000$ human face images, which is more complicated than the datasets in our other experiments. In addition to the estimation method introduced before, we incorporate a sparsity regularization ($\ell_1$ norm) on the Jacobian of the mixing function $f$, as required by (Zheng et al., 2022; Zheng & Zhang, 2023). Note that the identifiability theory in (Kong et al., 2022) does not require specific regularization during estimation if the task is not domain adaptation.

From Figs. 12, 13, and 14, it is evident that some concepts remain entangled and cannot be fully recovered. For instance, for the class "Age", concepts like "skin," "eye," and "face shape" are all entangled together, suggesting that assumptions in (Zheng et al., 2022; Zheng & Zhang, 2023; Kong et al., 2022) for component-wise identifiability may not be fully satisfied in this scenario. However, these class-related concepts can still be identified as a group, consistent with our theorem based on local or pairwise comparisons. This suggests that, even in complex scenarios where prior theories fail to guarantee identifiability due to assumption violations, our alternative identifiability framework based on pairwise comparisons may still provide an alternative theoretical basis for recovering class-related concepts collectively, even if they remain entangled. This sheds light on the necessity of our alternative identifiability guarantees in some complicated real-world scenarios.

