# OpenReview forum: "Provably Learning Concepts by Comparison"
_ICLR.cc/2025/Conference — Submitted to ICLR 2025_

### Official Review · Reviewer_Z5MD · 2024-11-02

**Soundness:** 3
**Presentation:** 1
**Contribution:** 2
**Rating:** 6
**Confidence:** 2

**Summary:**

This paper studies a new model of concept learning that works roughly as follows:
- In any given setting (e.g. classifying images of animals), we have $u$ possible classes, such as "cat", "dog", "mammal", "bird" (not necessarily mutually exclusive), and $n$ possible real-valued concepts, such as "furriness", "number of legs", "age", etc. Of these, $n_A$ concepts are class-dependent (e.g. "furriness", "number of legs"), and the rest are not (e.g. "age").
- A binary "structure" matrix $M \in \\{0, 1\\}^{n_A \times u}$ describes which concepts are related to which classes.
- Data is generated as follows: we first generate a class vector $c \in \\{0, 1\\}^u$ (e.g., "cat" + "mammal"), then we generate a latent concept vector $z$ based on $c$ (via a distribution $p(z|c)$ that follows the dependence structure given by $M$), and finally generate an observed vector $x$ based on the concept vector $z$.
- It is important that no other parametric assumptions are made.
- The goal is to identify $z$ based only on the observed data $x$.

Under this data-generating process, the main results are identifiability results:
- Under some technical assumptions that essentially encode a kind of diversity and identifiability in the structure matrix (intuitively, for each concept $z_i$, there must be a set of classes such that $z_i$ is unique to one of those classes --- in a sense, each concept has a unique "class signature"), the concept vector $z$ is identifiable from the data up to permutation and invertible transformation (Theorem 2).
- Extensions / variants of this result are also given (e.g. for pairwise comparison of classes, etc).
- Some experimental results are also given.

**Strengths:**

**Disclaimer:** while my background is in ML theory, I have very limited familiarity with this particular area (which I see as causal discovery / structure learning), and my perspective should be taken as that of a relative outsider.

This work tackles a broad and ambitious question in "concept learning": what are the mildest structural assumptions under which we can reasonably hope to identify underlying concepts from purely observed data? To do so, the paper introduces a rather generic non-parametric data-generating process and shows basic identifiability results under certain technical assumptions.

Overall the paper scores well on ambition and originality, as this is certainly an important question to investigate, and it appears from the literature that prior work had not tackled the question in quite such generality (to my limited knowledge). Thus a nice contribution of the paper is the definition of the model and the approach to the question. It is also nice that the authors obtain results with very few parametric assumptions on the data-generating process. The result is likely of technical interest to an audience in causality and related areas.

**Weaknesses:**

(See disclaimer above.)

By far the biggest weakness of the paper is that it is written in a very obscure and technical fashion, and is IMO very hard to read and understand for anyone not working in causal discovery, structure learning or related areas. Remarkably for a paper that claims to study such a broad and basic question, essentially no examples whatsoever are given of how the main data-generating process is instantiated in real-world settings. In fact it took me a while even to extract enough understanding to write the summary above.

In general the paper seems jump between very technical statements and voluminous but vague exposition. For example, the main theorems are all quite technical and tricky to parse. The few examples given are entirely synthetic and hard to interpret. The notation is quite dense. See also some concrete questions below.

I also have a lurking suspicion that the framework is so generic and the assumptions so strong that the main result is sort of tautological and primarily a mathematical exercise. Intuitively it seems quite reasonable to expect diversity assumptions of this type to lead to identifiability. Perhaps the authors could do a better job contextualizing what makes the result surprising and notable.

Overall it is not clear who the intended audience really is. As interesting as the results may be, they certainly do not seem accessible to a broad machine learning audience like at ICLR. While the authors have placed it in the primary area of "interpretability and explainable AI", I am quite sure it is out of place here, at least as written. In fact I am not sure why it has not been framed more squarely as causal discovery or structure learning, or indeed submitted to a specialized technical venue or journal in that field.

**Questions:**

- I strongly recommend that the authors discuss real-world examples of how the data-generating process may be instantiated (e.g. for image classification in the context of Fashion-MNIST or AnimalFace, but also others). This includes clearly specifying what the classes, concepts, and observed values are in each example.
- A high-level modeling question I am unclear on is why there is a distinction between classes and concepts. Any given observed vector is implicitly associated both with a latent vector of classes and a latent vector of concepts. Both appear to be unobserved --- unless the classes are indeed observed and this is the difference? If not it is not clear why we want to model them separately at all, and how they are playing functionally different roles.
- What is the type of each class $c_i$? What space does it take values in? This does not seem clearly specified anywhere and is very confusing. The entire meaning of the partial derivative $D_c g$ rests on this.
- Theorem 1 refers to "estimated latent concepts" --- what is the estimation procedure? If it is an identifiability result, why do we need to refer to estimated concepts at all?
- It would really great if the notation could be lightened in any way possible. The array slicing notation $D_{:n_A, :}$ in particular adds a lot of visual noise and I am not sure why it is needed if $g$ anyway maps to $\\mathbb{R}^{n_A}$. At least some shorthand could be introduced.

---

> ### Author Response · Authors · 2024-11-15
> **We are profoundly thankful for your insightful feedback (1/3)**
>
> We are profoundly thankful for your insightful feedback and the time you have dedicated to reviewing our work. Your suggestions have been invaluable in helping us improve the manuscript. In response, we have carefully revised the presentation, with particular focus on enhancing the clarity and accessibility of our theoretical results. Please kindly find our point-by-point responses below.
>
>
> **Q1:** Real-world examples of how the data-generating process may be instantiated.
>
> **A1:** Thank you very much for your constructive suggestion. We fully agree that providing real-world examples of the generative process will greatly aid readers in understanding the theory. While we have illustrated different types of variables throughout the manuscript (e.g., Fig. 1, lines 86, 229-236), your suggestion made us realize that a standalone, complete example would make it easier for readers to follow. Therefore, we have added the following example in the preliminaries:
>
> > “ (Line 119) Example 1. Consider images of animals in an aquarium, where the observed variables $\mathbf{x}$ represent image pixels. The different animal types (e.g., “shark” and “turtle”) correspond to classes $c$. Class-dependent concepts might include attributes like “predator,” “sleek body,” and “ocean” (see, e.g., Fig. 1), while class-independent concepts could be “lighting” and “position.” The hidden generative process of each image depends on all of these concepts, though only some are specific to each class.”
>
> Thank you again for this valuable input, which has helped us make the theoretical framework more accessible to readers.
>
>
> **Q2:** Why is there a distinction between classes and concepts? Both appear to be unobserved --- unless the classes are indeed observed and this is the difference?
>
> **A2:** Great point. You are correct: in our framework, the classes are observed, while the concepts remain latent. We have clarified this distinction further in the preliminaries to make it clearer for readers. Meanwhile, the class variables serve as “higher-level” elements relative to the concepts, with each class associated with multiple underlying concepts. This naturally forms a compositional structure that may be helpful for tasks such as compositional generalization and novel class discovery.
>
>
> **Q3:** What is the type of class variable $\mathbf{c}$?
>
> **A3:** Thank you for reviewing the paper so carefully, and we apologize for any confusion. In the updated manuscript, we have clarified that $\mathbf{c}$ is a real-valued surrogate variable representing the object’s class. The partial derivative serves as a technical tool to capture the dependency structure between classes and concepts in the general nonlinear case. For practical cases involving only discrete values, a mask can be used to model this structure instead. In light of your constructive comment, we have emphasized this clarification in the updated manuscript to prevent potential confusion.
>
>
> **Q4:** Theorem 1 refers to "estimated latent concepts" --- what is the estimation procedure? If it is an identifiability result, why do we need to refer to estimated concepts at all?
>
> **A4:** Thanks for raising this point. Indeed, the work studies identifiability theory and thus is analogous to the estimator. The proposed theory does not specify a particular algorithm or estimator but rather addresses the conditions under which latent concepts can be identified from observations. In the identifiability literature, terms like "estimated latent concepts" or "estimated latent variables" are commonly used to describe the identifiability objective (e.g., [Hyvärinen & Morioka, 2016; Khemakhem et al., 2020; Sorrenson et al., 2020; Lachapelle et al., 2022; Hyvärinen et al., 2024]). As such, the specific estimation procedure is introduced later in the experimental setup. To clarify, we have added the following explanations next to Theorem 1 and in the preliminaries:
>
> > “(Line 193) It is worth noting that the identifiability theory remains agnostic to the choice of estimator, provided the marginal distributions of the observations are matched. The results demonstrate that for any pair of classes, the unique concepts specific to each class can be disentangled from the other concepts.”
>
> > “(Line 136) Since identifiability is one of the main concerns, the theory is agnostic to estimators and the goal is to fit the marginal distribution $p(\mathbf{x})$ with model (learner) $\hat{f}$ and estimated variables $\hat{\mathbf{z}}$ to achieve certain identifiability.”

---

> > ### Author Response · Authors · 2024-11-15
> > **We are profoundly thankful for your insightful feedback (2/3)**
> >
> > To provide a more comprehensive background on causal representation learning and identifiable latent variable models, we have also introduced definitions of related identifiability objectives in the preliminaries:
> >
> > > “(Line 138) We introduce several identifiability objectives (Hyvärinen & Morioka, 2017; Lachapelle et al., 2022; Zheng et al., 2022; Kong et al., 2022; Hyvärinen et al., 2024) that are common in the literature as follows:”
> >
> >
> > > “(Line 141) Definition 1 (Element-wise Identifiable). The set of latent variables $\mathbf{z} \subseteq \mathbb{R}^n$ are \textit{element-wise identifiable} if there exists an invertible function $h_i: \mathbb{R} \rightarrow \mathbb{R}$ and a permutation $\pi$ s.t. $\hat{\mathbf{z}}_i = h_i(\mathbf{z}\_{\pi(i)})$.”
> >
> >
> > > “(Line 143) Definition 2 (Subspace-wise Identifiable). The set of latent variables $\mathbf{z} \subseteq \mathbb{R}^n$ are \textit{subspace-wise identifiable} if there exists an invertible function $h: \mathbb{R}^n \rightarrow \mathbb{R}^n$ s.t. $\hat{\mathbf{z}} = h(\mathbf{z})$.”
> >
> >
> > Thank you again for your suggestion. We hope this added clarification will help make our theoretical framework more accessible to a broader audience. Please kindly let us know if you have any further feedback.
> >
> >
> > **Q5:** It would be really great if the notation could be lightened in any way possible
> >
> > **A5:** Thank you very much for this valuable suggestion. We truly appreciate the opportunity to make our work more accessible, and we fully agree that simplifying notation is key to improving readability. Given the theoretical density of our work, we have made targeted adjustments based on your feedback to make the notation as clear as possible.
> >
> > Beyond the initial submission’s simplifications—such as introducing shorthands, adding visual aids to clarify the problem setup, and providing a notation summary—we have also made the following updates in light of your suggestion:
> >
> > 1. Simplifying the array slicing whenever possible in both the main paper and appendix, for example
> > 	- “$\mathcal{D}\_{:n_A,i}$” -> “$\mathcal{D}\_{:,i}$”
> > 	- “$\mathcal{D}\_{:n_A,:}$” -> “$\mathcal{D}$”
> >
> > 2. In addition to the existing shorthands (e.g.,  $\mathcal{D}$ for $\operatorname{supp}(D_\mathbf{c} g)$), we have also changed the $(c,\theta, \epsilon)$ to $(c, \theta)$, where $\theta$ itself represents all other factors including potential noise.
> >
> > We hope these refinements make the notation more accessible, and we greatly appreciate your suggestion, which has helped us improve the readability of our work.
> >
> >
> > **Q6:** Perhaps the authors could do a better job contextualizing what makes the result surprising and notable.
> >
> > **A6:** Thank you very much for this helpful suggestion. We would like to contextualize our contributions from two key perspectives:
> >
> > **Related work.** As highlighted in the abstract and introduction, prior works on concept learning have primarily focused on empirical performance without offering theoretical guarantees. A few studies have explored the theoretical side, but these typically rely on specific parametric assumptions, such as linear relationships among all concepts or additivity in the generating process—assumptions that may not hold in many real-world settings. In contrast, our theoretical framework operates without these constraints, aiming to establish identifiability in a broad, nonparametric setting.
> >
> > **Techniques.** Since our goal is to identify which concepts can be reliably recovered in the most general settings, we extend beyond traditional parametric assumptions by leveraging insights from fundamental learning mechanisms. Specifically, we introduce a flexible identifiability framework based on the cognitive process of learning by comparison. For instance, given any two pairs of classes, our approach allows us to disentangle latent concepts that represent each class’s unique features.
> >
> > This is fundamentally and technically different from most prior works on the identifiability of latent variable models, even though the underlying problem is distinct. Specifically, existing works (see e.g., a recent survey [Hyvärinen et al., 2024]) focus on identifying all latent variables as a complete hidden representation and generally lack flexibility for localized insights. If any variables in the considered set do not satisfy their assumptions, these theories cannot guarantee any degree of identifiability. By contrast, we can always provide alternative identifiability guarantees as long as there exists diversity, even if only between two classes.

---

> > > ### Author Response · Authors · 2024-11-15
> > > **We are profoundly thankful for your insightful feedback (3/3)**
> > >
> > > In light of your great suggestion, we have further highlighted these distinctions in the updated manuscript, such as
> > >
> > > > “(Line 260) … Most latent variable identifiability works also face the same challenge dealing with partial assumption violation (Zheng et al., 2022; Kong et al., 2022; Zheng & Zhang, 2023, Hyvärinen et al., 2024). Unlike our local or even pair-wise identification strategy, these methods lack the flexibility to recover arbitrary parts of the hidden process in a localized manner.”
> > >
> > > We have also added further discussion comparing our strategy with other works on identifiability from additional perspectives, while most of these studies do not address concept learning. Some examples include:
> > >
> > > > “(Line 322) Different from various assumptions encouraging the sparsity of the structure in the literature (Rhodes & Lee, 2021; Moran et al., 2021; Zheng et al., 2022; Zheng & Zhang, 2023), our assumption only ensures necessary variability on the dependency structure and could also hold true with relatively dense connections. At the same time, we permit arbitrary structures between the class-dependent hidden concepts and the observed variables, while previous work has to assume a sparse structure on the generating process between latent and observed variables.”
> > >
> > > > “(Line 328) Additionally, another line of work on latent variable models requires $2n_A + 1$ distinct domains or classes to achieve latent variable identifiability (e.g., (Hyvärinen & Morioka, 2017; Khemakhem et al., 2020a; Kong et al., 2022; Hyvärinen et al., 2024)), a condition we do not impose.”
> > >
> > > We sincerely appreciate your constructive suggestion, which has greatly helped us improve the contextualization and clarity of our contributions. Please kindly let us know if there are any further ways we can enhance the manuscript.
> > >
> > > **Q7:** The intended audience for the paper is unclear. The primary area of "interpretability and explainable AI" may not be the best fit, and the work might align more closely with fields like causality.
> > >
> > > **A7:** Thank you very much for your feedback. We deeply appreciate your perspective on the intended audience and the paper’s alignment with relevant fields. Since the identifiability of the hidden data-generating process provides a principled way to uncovering the underlying structure of observational data, we initially selected explainability as the primary area. Specifically, identifying concepts allows us to reveal the hidden generative factors and the latent compositional structures that underlie different classes of observations. The field of interpretable and explainable AI is broad, fundamental, and critically important. We aim to contribute to this field from a novel perspective and learn from the valuable feedback provided by the community.
> > >
> > > At the same time, we agree that some of our results are also relevant to fields like causality, as both areas share the overarching goal of exploring and understanding the hidden mechanism. To reflect this, we have further highlighted additional fields that may find our results valuable in the updated manuscript:
> > >
> > > > “(Line 375) As a result, part of the proposed results might also be of independent interest to other fields such as disentanglement (Hyvärinen et al., 2024), causal representation learning (Schölkopf et al., 2021), object-centric learning (Mansouri et al., 2024), compositional generalization (Du & Kaelbling, 2024), and causal structure learning (Spirtes et al., 2000).”
> > >
> > > Thank you once again for your constructive input. Please feel free to let us know if you have any further feedback or suggestions.
> > >
> > > ---
> > >
> > > References:
> > >
> > > [1] Hyvärinen & Morioka, Unsupervised feature extraction by time-contrastive learning and
> > > nonlinear ICA, NeurIPS 2016
> > >
> > > [2] Khemakhem et al., variational autoencoders and nonlinear ICA: A unifying framework, AISTATS 2020
> > >
> > > [3] Sorrenson et al., Disentanglement by nonlinear ICA with general incompressible-flow networks (GIN), ICLR 2020
> > >
> > > [4] Lachapelle et al., Disentanglement via mechanism sparsity regularization: A new principle for nonlinear ICA, CLeaR 2022
> > >
> > > [5] Hyvärinen et al., Identifiability of latent-variable and structural-equation models: from linear to nonlinear, Annals of the Institute of Statistical Mathematics, 2024

---

> ### Author Response · Authors · 2024-11-25
> **We are grateful for your time and thoughtful feedback**
>
> Dear Reviewer Z5MD,
>
> Thank you very much for taking the time to provide your thoughtful feedback. We hope our responses have addressed your concerns. As the discussion period nears its end, we warmly welcome any further input you may have and would be more than happy to engage in discussions.
>
> Best wishes,
>
> Authors of Submission8461

---

> > ### Comment · Reviewer_Z5MD · 2024-11-25
> >
> > Thank you for your responses and updates to the manuscript. While they help, I continue to feel the paper is borderline in its current form, and will keep my rating.

---

> > > ### Author Response · Authors · 2024-11-25
> > > **Thanks so much for your prompt response**
> > >
> > > Dear Reviewer Z5MD,
> > >
> > > Thank you for your follow-up comment and for taking the time to engage with our responses and updates. We deeply value your feedback, which has been instrumental in improving the manuscript.
> > >
> > > As you kindly noted, our work addresses **“a broad and ambitious question in concept learning”** that **“prior work had not tackled in quite such generality”** and **“scores well on ambition and originality”**. This has further motivated us to make every effort to share our theory with the community in a timely manner. To this end, we have worked diligently to address any potential confusion or misunderstanding arising from the complex nature of the problem and believe we have clarified all key points of concern. Specifically, we have worked around the clock to provide complete and detailed responses as soon as possible (*within two days*) to ensure more time for discussion. If there are any specific aspects you feel we haven’t fully addressed, we would be most grateful for the opportunity to provide further clarifications or revisions during the discussion period to better meet your expectations.
> > >
> > > We sincerely appreciate your time and effort throughout the review process and remain committed to ensuring this work has the maximum possible impact. We genuinely hope you might reconsider your assessment and give our submission another chance. Please don’t hesitate to reach out with any additional questions or feedback.
> > >
> > > Best wishes,
> > >
> > > Authors of Submission8461

---

> > > > ### Comment · Reviewer_Z5MD · 2024-11-28
> > > >
> > > > Here are the reasons why I continue to feel uncomfortable endorsing this paper, after taking another close look at the new changes:
> > > > 1. The definitions of identifiability continue to seem obscure and ill-formed to me. I'm not familiar enough with the causality / concept learning literature to know if these terms are used differently there, but in classical statistics the definition of identifiability is fundamentally a property of _latent variables / parameters_ and _observed variables_ --- _not_ estimated parameters in any way. It refers to the limitations of any estimation procedure even in principle. The traditional and strongest form is that the distribution of observed variables is a one-to-one function of the latent variables / parameters. So to be extremely concrete, the way I would expect e.g. Def 2 to look is roughly as follows: "a set of latent variables $z$ is subspace-wise identifiaible if there exists an invertible function $h$ such that for any two $z_1$ and $z_2$ that result in the same distribution of observed variables,  $z_1 = h(z_2)$". The point being that it must formalize the extent to which the true latents are uniquely identifiable given the observed variables --- regardless of estimation procedure.
> > > > 2. As a related note, I don't know why Def 2 is called "subspace-wise" invertibility at all. The function $h$ could be highly nonlinear, and I don't see where subspaces enter the picture at all.
> > > > 3. For these and other such reasons, I just feel the paper is still hard to understand and has not sharpened its message for a general audience. This is not to say the work is not worthwhile, but I feel the paper needs a more substantive reworking. Also ultimately note that mine is not a high-confidence review owing to lack of close familiarity with this area.

---

> > > > > ### Author Response · Authors · 2024-11-28
> > > > > **We sincerely appreciate your thoughtful feedback and the opportunity to provide clarification**
> > > > >
> > > > > Dear Reviewer Z5MD,
> > > > >
> > > > > We sincerely appreciate your thoughtful feedback and the opportunity to provide clarification. We are delighted to see that we share the same understanding, and the confusion can be promptly resolved.
> > > > >
> > > > > The definition of subspace-wise identifiability (also known as block-wise identifiability) is **widely used in the literature on causal representation learning and latent variable model identifiability**. For instance, it appears in Defn. 4.1 in [Von Kügelgen et al., 2021], Thm. 4.2 in [Kong et al., 2022], Thm. 1 in [Li et al., 2023], and Defn. 2.3 in [Yao et al. 2024]. It means that the estimated latent variable **subset** contains ***all*** and ***only*** the information about the corresponding ground-truth latent variable **subset**. In our theorem regarding subspace-wise identifiability (Theorem 2), subspace-wise identifiability of $\mathbf{z}_B$ ensures that $\mathbf{z}_B$ is disentangled from $\mathbf{z}_A$, i.e., contains no information from $\mathbf{z}_A$, after estimation (matching the observed distributions). Specifically, proving $\hat{\mathbf{z}}_B = h(\mathbf{z}_B)$ guarantees disentanglement between the subspaces $\mathbf{z}_A$ and $\mathbf{z}_B$, where $\mathbf{z} = [\mathbf{z}_A, \mathbf{z}_B]$.
> > > > >
> > > > > In light of your questions, we have added the following highlight immediately after Defn. 2 (in purple) to avoid any potential confusion for general audiences:
> > > > >
> > > > > > “(Line 147) It might be worth noting that the subspace-wise identifiability implies the disentanglement between subsets of latent variables. For instance, if $\mathbf{z}_B$ is subspace-wise identifiable, then $\mathbf{z}_B$ will not contain any information from $\mathbf{z}_A$ after estimation. The subspace-wise identifiability is commonly used in the literature (Von Kügelgen et al., 2021; Kong et al., 2022; Li et al., 2023; Yao et al. 2024)”
> > > > >
> > > > > At the same time, in the previous updates, we have added the following sentences to highlight that the identifiability is estimator-agnostic. Therefore, we believe we are on the same page regarding the definition of identifiability, and the potential confusion could be addressed by these clarifications.
> > > > >
> > > > > > “(Line 136) Since the considered problem is identifiability, the theory is agnostic to estimators and the goal is to fit the marginal distribution $p(\mathbf{x})$ with model (learner) $\hat{f}$ and estimated variables $\hat{\mathbf{z}}$ to achieve certain identifiability.”
> > > > >
> > > > > Moreover, we have conducted more synthetic and real-world experiments to further support our theoretical results (Appendix C.2). Given the significant empirical successes in the literature of concept learning but limited theoretical exploration, we really hope to make every effort to provide as many insights as possible to the community.
> > > > >
> > > > > Thank you once again for your valuable time and thoughtful efforts. It is truly an honor for us to have your support in improving the manuscript and making it more accessible to a broader audience. Please do not hesitate to reach out with any further questions or suggestions—we would be most grateful for the opportunity to address them.
> > > > >
> > > > >
> > > > >
> > > > > With sincere gratitude,
> > > > >
> > > > > Authors of Submission8461
> > > > >
> > > > > ---
> > > > >
> > > > > References:
> > > > >
> > > > > [1] Von Kügelgen et al., Self-supervised learning with data augmentations provably isolates content from style, NeurIPS 2021
> > > > >
> > > > > [2] Kong et al., Partial identifiability for domain adaptation, ICML 2022
> > > > >
> > > > > [3] Li et al., Subspace identification for multi-source domain adaptation, NeurIPS 2023
> > > > >
> > > > > [4] Yao et al., Multi-View Causal Representation Learning with Partial Observability, ICLR 2024

---

> > > > > > ### Comment · Reviewer_Z5MD · 2024-12-01
> > > > > >
> > > > > > Appreciate the clarification. I still find this formulation of identifiability unusual, but I will let the matter rest if that is how it is used in the literature. Incidentally I would recommend using the term "block-wise identifiability" if there is already precedent for this particular definition.
> > > > > >
> > > > > > I will increase my score to 6.

---

> > > > > > > ### Author Response · Authors · 2024-12-01
> > > > > > > **We really appreciate your positive feedback and constructive suggestions**
> > > > > > >
> > > > > > > Dear Reviewer Z5MD,
> > > > > > >
> > > > > > > We really appreciate your positive feedback and constructive suggestions. While both "subspace-wise" and "block-wise" have been used in prior literature, we are happy to adopt the term 'block-wise identifiability' in the manuscript if it aligns better with clarity and intuition. Thank you once again for your valuable time and efforts in helping us refine and improve our work.
> > > > > > >
> > > > > > > Many thanks,
> > > > > > >
> > > > > > > Authors of Submission8461

---

### Official Review · Reviewer_omEm · 2024-11-03

**Soundness:** 2
**Presentation:** 2
**Contribution:** 2
**Rating:** 3
**Confidence:** 2

**Summary:**

This paper models and studies the problem of concept learning. Suppose that we are given observations $x$s from $k$ classes that are generated based on their concept vectors $z$, furthermore, each class has a set of associated indices of the concept vectors and a concept vector is generated according to the class, we want to recover the corresponding set of concept indices for each class. The paper tries to build a theoretic framework that can explain under certain assumptions, which kind of concept can be recovered. Besides the theoretic framework, the paper also performs experiments on both synthetic and real-world datasets based on their theoretic framework.

**Strengths:**

Concept learning is an important topic in machine learning. This paper builds a mathematical framework to help understand what concepts can be reliably recovered. Experiments are also provided to help justify the theory in this paper.

**Weaknesses:**

1.
The main weakness of the paper is its presentation. I list several issues about the presentation as follows.

The definition of the problem is not complete in the preliminary section. In the preliminary section, the paper only defines how observations are generated but does not give a clear definition of what a learner is expected to output and how to measure the performance of a learner.
Furthermore, the statements of the main theorems are not mathematically formal enough. For example, the statement of Theorem 1 mentions "estimated latent concepts"
> the estimated latent concepts for the set difference $\hat{z}_{\pi(A_i \backslash A_j)}$...,

I understand that the estimated latent concepts should be the output of a learner. However, before the statement of Theorem 1, the paper does not define the definition of estimated latent concepts. It is confusing what algorithm/estimator a learner uses to get such an estimation. Another example is the statement of Theorem 2. In the statement of Theorem 2, the paper mentions
>$z_B$ is identifiable up to a subspace-wise invertible transformation.

I think subspace-wise invertible transformation is an informal mathematical term. Furthermore, as the definition of identifiable is also not formally presented in the preliminary section, it is unclear the exact result that the theorem would like to convey.

2. I think some of the assumptions made by the paper are not natural enough. For example, in equation (3), the paper assumes that the entries of the class-dependent concepts $z_A$ are conditionally independent. This seems to be a very strong assumption, but the paper does not provide natural examples to justify that the assumption is reasonable.

**Questions:**

1.
In the preliminary section, the classes are defined as a vector $c=(c_1,\dots,c_u)$. It is not very clear to me whether the class vector $c$ can take continuous real values or can only take discrete values. If $c$ can only take discrete values, then in line 124, it is not reasonable to say one can take the partial derivative of $g$ with respect to $c$. Can you help clarify this?

2.
In line 320, it says the distributional assumption in Theorem 2 necessitates the existence of at least two classes with differing conditional distributions and is highly likely to be satisfied. But it seems that the distribution assumption is much stronger because it requires the probability mass for the probabilities to be unequal for any subset $A_z$ that is not a product set of $B_{z_B} \times z_A$. Can you provide any real-world example that satisfies the assumption?

---

> ### Author Response · Authors · 2024-11-15
> **We deeply appreciate the valuable insights you have shared (1/3)**
>
> We deeply appreciate the valuable insights you have shared, which have helped us enhance the quality and clarity of the manuscript. In light of your great feedback, we have carefully revised the manuscript, adding further clarifications and explanations. Please kindly find our detailed response below.
>
>
> **Q1:** The main weakness of the paper is its presentation. I list several issues about the presentation as follows.
>
> **A1:** Thank you very much for your comments, which have nicely inspired us to rewrite some paragraphs to make the messages more explicit for a broader audience. We hope your concerns have been properly addressed, and would be delighted if you find the key messages we aim to deliver interesting and exciting. We provide detailed responses to each of the comments as well as the corresponding updates as follows:
>
> - **Q1.1:** Lack of a clear definition of what a learner is expected to output and how to measure the performance of a learner.
>
> - **A1.1:** Thank you for highlighting this point—we truly value the opportunity to clarify our focus. As mentioned in line 136 in the preliminaries, $\hat{z}$ denotes estimated concepts, which represents the output of a learner. Our work centers on identifiability theory, specifically on establishing the conditions under which the latent data-generating process (model) can be identified (up to a certain indeterminacy) from observations. Since our study is grounded in theoretical foundations rather than specific implementation, it is designed to be learner-agnostic and does not specify a performance measure. The specific estimation procedure is thus described later in the experimental setup. In light of the comment, we have also added the following sentence in the preliminary:
>
>   > “(Line 137) … the goal is to fit the marginal distribution $p(\mathbf{x})$ with model (learner) $\hat{f}$ and estimated variables $\hat{\mathbf{z}}$ to achieve certain identifiability.”
>
>   Furthermore, we recognize that further emphasizing this scope could be beneficial and have added clarifications throughout the updated manuscript. For example:
>
>
>   > “(Line 193) It is worth noting that the identifiability theory remains agnostic to the choice of estimator, provided the marginal distributions of the observations are matched. The results demonstrate that for any pair of classes, the unique concepts specific to each class can be disentangled from the other concepts.”
>
>   We hope this clarification helps make our task clearer. Thank you once again for your feedback.
>
> - **Q1.2:** It is confusing what algorithm/estimator a learner uses to get such an estimation.
>
> - **A1.2:** Thank you for raising this question. As mentioned in our response to Q1.1, our work focuses on the identifiability theory, which is intentionally learner-agnostic. The theory does not specify a particular algorithm or estimator but rather addresses the conditions under which latent concepts can be identified from observations. Therefore, the specific estimation procedure is included in the experimental setup, which is basically a MLE. In the identifiability literature, terms like "estimated latent concepts" or "estimated latent variables" are commonly used to describe the identifiability objective (e.g., [Hyvärinen & Morioka, 2016; Khemakhem et al., 2020; Sorrenson et al., 2020; Lachapelle et al., 2022; Hyvärinen et al., 2024]). We appreciate your feedback and have further highlighted it in the updated manuscript:
>
>   > “(Line 136) Since identifiability is one of the main concerns, the theory is agnostic to estimators and the goal is to fit the marginal distribution $p(\mathbf{x})$ with model (learner) $\hat{f}$ and estimated variables $\hat{\mathbf{z}}$ to achieve certain identifiability.”

---

> ### Author Response · Authors · 2024-11-15
> **We deeply appreciate the valuable insights you have shared (2/3)**
>
> - **Q1.3:** What is the meaning of "identifiable up to a subspace-wise invertible transformation"?
>
> - **A1.3:** Thanks for the comment. By "identifiable up to a subspace-wise invertible transformation," we mean that variables can be identified up to an invertible mapping between subspaces instead of the entire space. This objective is prevalent in the identifiability literature [Kong et al., 2022; Lachapelle et al., 2022; Li et al., 2023; Zheng & Zhang et al., 2023; Hyvärinen et al., 2024], where it is sometimes referred to as "subspace-wise" or "block-wise" identifiability.
>
>   To avoid any potential confusion, we have added the following definitions in the preliminary section for a more comprehensive coverage of the related background:
>
>   > “(Line 138) We introduce several identifiability objectives (Hyvärinen & Morioka, 2017; Lachapelle et al., 2022; Zheng et al., 2022; Kong et al., 2022; Hyvärinen et al., 2024) that are common in the literature as follows:”
>
>
>   > “(Line 141) Definition 1 (Element-wise Identifiable). The set of latent variables $\mathbf{z} \subseteq \mathbb{R}^n$ are *element-wise identifiable* if there exists an invertible function $h_i: \mathbb{R} \rightarrow \mathbb{R}$ and a permutation $\pi$ s.t. $\hat{\mathbf{z}}_i = h_i(\mathbf{z}\_{\pi(i)})$.”
>
>
>   > “(Line 143) Definition 2 (Subspace-wise Identifiable). The set of latent variables $\mathbf{z} \subseteq \mathbb{R}^n$ are *subspace-wise identifiable* if there exists an invertible function $h: \mathbb{R}^n \rightarrow \mathbb{R}^n$ s.t. $\hat{\mathbf{z}} = h(\mathbf{z})$.”
>
>
>   To further enhance clarity, we also reference these definitions in our theorems to assist readers who may be less familiar with this topic. Thank you again for your helpful feedback, which has allowed us to make these details more accessible to a broader audience.
>
>
> **Q2:** Assumption of conditional independence seems to be strong.
>
> **A2:** Thank you for your feedback. In our setting, the relationship between class and concepts makes conditional independence a natural formulation of the problem. For instance, although the concepts "wings" and "feathers" are semantically dependent, they become conditionally independent when given the class variable "bird." This highlights a meaningful form of modularity that aligns with the structure of class-concept relationships in our framework.
>
> Intuitively, some form of modularity in latent variables is essential to recover them individually, and conditional independence is a well-established formulation, frequently used in statistical frameworks like independent component analysis. It is a common assumption in prior work on the identifiability of latent variable models [Hyvärinen & Morioka, 2016;
> Khemakhem et al., 2020; Sorrenson et al., 2020; Lachapelle et al., 2022; Hyvärinen et al., 2024]. Even in rare cases the latent variables are actually conditionally dependent, we can still make use of some other additional information, such as multiple domains [Zhang et al., 2024].
>
> Moreover, it may be helpful to note that our constraints are, in fact, less restrictive than those in prior works on identifiable concept learning such as [Rajendran et al., 2024]. Whereas previous studies often assume (conditional) independence across all concept variables $\mathbf{z}$, our work allows for arbitrary dependencies among class-independent concepts, thereby accommodating more general scenarios.
>
> In response to your excellent suggestion, we have incorporated examples like the following into the paper to illustrate this:
>
> > “(Line 153) The conditional independence provides a form of modularity commonly adopted in prior work on identifiable latent variable models (Hyvärinen & Morioka, 2016;
> Khemakhem et al., 2020; Sorrenson et al., 2020; Lachapelle et al., 2022; Hyvärinen et al., 2024). It may be particularly suitable in our class-concept framework; for example, while the concepts 'wings' and 'feathers' are related, they become conditionally independent given the class variable 'bird.'”
>
> Thank you once again for your thoughtful feedback, which has helped us enhance the clarity and contextualization of this assumption.
>
>
>
>
> **Q3:** Can the class vector $\mathbf{c}$ take continuous real values?
>
> **A3:** Thanks for this observation. You are absolutely correct; the class vector $\mathbf{c}$ can indeed take continuous values. We have highlighted this in the updated manuscript to improve clarity. The derivative serves as a technical tool to represent the dependency structure between classes and concepts in the general nonlinear setting. In practical scenarios where only discrete values are available, a mask can be used to effectively represent the structure. We appreciate your feedback, which has helped us clarify this aspect of the paper.

---

> > ### Author Response · Authors · 2024-11-15
> > **We deeply appreciate the valuable insights you have shared (3/3)**
> >
> > **Q4:** More discussion on the distributional assumption in Theorem 2.
> >
> > **A4:** Thank you very much for raising this point. We realize that our initial explanation of the distributional assumption may have inadvertently made it appear stronger than intended, leaving some room for potential misunderstanding. We have clarified this in the revised manuscript to address this:
> >
> > > “(Line 283) … there exist two values of $\mathbf{c}$, i.e., $c^{(k)}$ and $c^{(v)}$ (which may vary across different $A_{\mathbf{z}}$)”
> >
> > Specifically, for any $A_{\mathbf{z}}$, although we require the probabilistic difference between classes $c^{(k)}$ and $c^{(v)}$, these two classes are not fixed and may change over different $A_{\mathbf{z}}$, providing great flexibility. This assumption is common in recent identifiability works [Kong et al., 2022; Xie et al., 2022; Zheng & Zhang, 2023] and was originally introduced in [Kong et al., 2022]. Intuitively, this assumption holds as long as the probability distributions for each class are not overly similar, a condition empirically validated by Kong et al. For example, in a zoo setting, it would be highly unlikely for all animal species to exhibit identical behaviors and features, thereby satisfying this type of probabilistic variation across classes.
> >
> > To further enhance clarity, we have added the following example in the manuscript:
> >
> > > “(Line 353) Importantly, these two classes may vary across different $A_{\mathbf{z}}$. Therefore, this assumption is highly likely to be satisfied in real-world scenarios, as it is virtually impossible for the measures corresponding to \textit{all} classes (e.g., all kinds of animals in a zoo) to be almost identical.”
> >
> > We sincerely appreciate your feedback, which has been instrumental in improving the clarity and accessibility of our work. Please kindly let us know if there are any further suggestions.
> >
> > ---
> >
> > References:
> >
> > [1] Hyvärinen & Morioka, Unsupervised feature extraction by time-contrastive learning and
> > nonlinear ICA, NeurIPS 2016
> >
> > [2] Khemakhem et al., variational autoencoders and nonlinear ICA: A unifying framework, AISTATS 2020
> >
> > [3] Sorrenson et al., Disentanglement by nonlinear ICA with general incompressible-flow networks (GIN), ICLR 2020
> >
> > [4] Lachapelle et al., Disentanglement via mechanism sparsity regularization: A new principle for nonlinear ICA, CLeaR 2022
> >
> > [5] Hyvärinen et al., Identifiability of latent-variable and structural-equation models: from linear to nonlinear, Annals of the Institute of Statistical Mathematics, 2024
> >
> > [6] Kong et al., Partial identifiability for domain adaptation, ICML 2022
> >
> > [7] Li et al., Subspace identification for multi-source domain adaptation, NeurIPS 2023
> >
> > [8] Zheng & Zhang, Generalizing nonlinear ICA beyond structural sparsity, NeurIPS 2023
> >
> > [9] Zhang et al., Causal representation learning from multiple distributions: A general setting, ICML 2024
> >
> > [10] Rajendran et al., Learning interpretable concepts: Unifying causal representation learning and foundation models, NeurIPS 2024

---

> > > ### Author Response · Authors · 2024-11-25
> > > **Thank you sincerely for your time and valuable suggestions**
> > >
> > > Dear Reviewer omEm,
> > >
> > > Thank you very much for your time and thoughtful suggestions. We hope we have sufficiently addressed your concerns. As the discussion period is coming to a close, please kindly let us know if you have any further feedback. We would be more than happy to continue the discussion with you.
> > >
> > > Best wishes,
> > >
> > > Authors of Submission8461

---

> ### Author Response · Authors · 2024-11-30
> **Could you please kindly let us know if the concerns have been properly addressed?**
>
> Dear Reviewer omEm,
>
> Thank you sincerely for your time and thoughtful feedback. Could you please kindly let us know if the concerns have been properly addressed? We believe that the main confusions have been clarified. Specifically, they are related to the **common practice** in the **identifiability literature**:
>
>   - Identifiability theory is estimator-agnostic, which is standard in the literature. In the updated manuscript, we have added clarifications and highlights throughout the paper to avoid potential confusion;
>
>   - Subspace-wise identifiability is widely used in the literature. In the updated manuscript, we have added relevant definitions as background in the preliminary;
>
>   - Conditional independence is a common assumption in previous theories, which we have further relaxed. In the updated manuscript, we have added further highlights on it.
>
> To prevent any potential misunderstanding, we have incorporated **additional highlights** and **expanded background definitions** in the revised version, following your excellent suggestions.
>
> Moreover, we have conducted **new synthetic and real-world experiments** to further support our theoretical results (Appendix C.2). Given the significant empirical successes in the literature of concept learning but the limited theoretical exploration, we are committed to providing as many insights as possible to benefit the community.
>
> If you have any further questions or additional feedback, we would be grateful to hear from you.
>
> Thank you again for your efforts and for contributing to the review process.
>
> Best regards,
>
> Authors of Submission8461

---

> > ### Author Response · Authors · 2024-12-02
> > **Kindly Requesting Feedback Before Discussion Concludes Today**
> >
> > Dear Reviewer omEm,
> >
> > Thank you very much for your efforts devoted to reviewing our manuscript. As mentioned in our detailed responses, we believe the concerns have been adequately addressed based on additional clarification and factual evidence. We have also conducted new experiments to further support the theory.
> >
> > Please note that the **deadline is approaching (*today*)**, and we would greatly appreciate it if you could kindly share any further feedback or update the rating accordingly.
> >
> > Thanks again for your time and consideration.
> >
> > Best wishes,
> >
> > Authors of Submission8461

---

> > > ### Author Response · Authors · 2024-12-03
> > > **Apologies for the repeated reminders; Discussion ends in four hours**
> > >
> > > Dear Reviewer omEm,
> > >
> > > Apologies for the repeated reminders. As the discussion period is approaching its end in approximately **four hours** and we have not yet heard from you, we are sincerely and eagerly looking forward to your feedback. We believe that our detailed responses have adequately addressed your concerns. We fully understand and appreciate your busy schedule and would be deeply grateful if you could kindly share any further feedback or update the rating accordingly. Thank you very much for your time and effort.
> > >
> > > With appreciation,
> > >
> > > Authors of Submission8461

---

### Official Review · Reviewer_NXKX · 2024-11-09

**Soundness:** 2
**Presentation:** 3
**Contribution:** 3
**Rating:** 8
**Confidence:** 2

**Summary:**

The paper studies learning of a certain type of non-parametric hidden variable model. Theorems are given showing statistical identifiability, and some experiments are preformed.

The family of non-parametric hidden variable models is described on pp 2-3. In brief:
- There are concept-dependent hidden variables $Z_A$ and concept-independent hidden variables $Z_B$. The distributions of $Z_A$ and $Z_B$ are not constrained to come from a specific distribution such as Gaussian. The concept-dependent variables $Z_A$ depend on the class $c$ of a given datapoint.
- The hidden variables $(Z_A, Z_B)$ are mapped to the observed variable $X$ via some smooth function $g$.
- It is further assumed that the Jacobian of $g$ satisfies certain full-rank assumptions in order to ensure an ability to recover the model based on observations.
- Some sparsity assumptions are made on how many concept-dependent hidden variables correspond to each class $c$.
- A "structural diversity" assumption is made stating roughly that each class has at least one hidden variable unique to this class and not shared with other classes.

The paper proves theorems that under the assumption listed above it is possible to recover the model (up to some transformation).

Experiments are performed on synthetic datasets, as well as Fashion-MNIST, EMNIST, AnimalFace, and Flower102 datasets and model is fitted using a regularized maximum-likelihood method. Robustness of the recovered concept is tested by comparing the same concept across different angles and environments.

The paper also compares the performance of regularized maximum-likelihood method that encorporates the aforementioned assumptions vs the base method that does not incorporate the assumptions above. It is shown that the former achieves a better performance in terms of Mean Correlation Coefficient (MCC).

**Strengths:**

- Non-parametric models of the type studied in this work have their uses and can be important if one is working with relatively low-dimensional data and needs high degree of interpretability.
- The paper is carefully written and includes an appendix listing all notation, figures are well-made.

**Weaknesses:**

- The paper does clearly articulate how exactly it improves on a number of closely-related papers such as [Zheng et al., 2022], [Zheng & Zhang, 2023] and [Kong et al. 2022] among others. These works study non-parametric models that look quite similar to the ones given in this work in terms of assumptions on the Jacobian and structural sparsity. They also use very similar mathematical tools to analyze identifyability. The paper does not articulate clearly what exactly it believes are some specific shortcomings of these earlier works, and how the current work overcomes such shortcomings. I think it is important that this is done, and it is also convincingly argued that the current paper indeed overcomes such limitations. (Update: added experiments partially address this objections, I raise my score but lower the confidence score)

**Questions:**

(See above)

---

> ### Author Response · Authors · 2024-11-15
> **We are genuinely grateful for your insightful comments (1/2)**
>
> We are genuinely grateful for the time you have dedicated and for your insightful comments. In response, we have added several new discussions in the updated manuscript, with a particular focus on comparisons to these prior works. Please find our detailed response below:
>
> **Q1:** Detailed comparison with earlier works such as [Zheng et al., 2022; Zheng & Zhang, 2023; Kong et al. 2024].
>
> **A1:** Thank you very much for connecting these seemingly different domains. Since both lines of research explore latent variable models, they indeed share insights toward building identifiability results. As mentioned in the discussion of implications (e.g., lines 372-377), our results might also be of theoretical interest to fields such as disentanglement and causal representation learning:
>
> > “(Line 372) Despite being one of the essential pieces on learning the hidden concepts, our proposed theory also sheds light on understanding the latent variable models without additional knowledge, since the formulation is just based on the basic generating process between latent and observed variables. As a result, part of the proposed results might also be of independent interest to other fields such as  disentanglement (Hyvärinen et al., 2024), causal representation learning (Schölkopf et al., 2021), object-centric learning (Mansouri et al., 2024), compositional generalization (Du & Kaelbling, 2024), and causal structure learning (Spirtes et al., 2000).”
>
> At the same time, there are some key distinctions:
>
> 1. While [Zheng et al., 2022; Zheng & Zhang, 2023; Kong et al., 2024] also consider the identifiability of latent variables, our problem of concept learning differs from theirs. As highlighted in line 63, our work aims to answer the question of which concepts can be reliably recovered in the most general case. Our primary identifiability result is rooted in the cognitive process of learning by comparison: given any pair of classes, the unique concepts corresponding to each class can be identified. This forms the foundation of our local identifiability based on pairwise comparisons and extends to more general cases, including global identifiability. For example, given any two pairs of classes, we can always disentangle latent concepts that represent each class’s unique features.
>
> 	In contrast, [Zheng et al., 2022; Zheng & Zhang, 2023; Kong et al., 2022] focus on identifying all latent variables as a complete hidden representation and lack the flexibility to provide insights in a localized manner. If any variables in the considered set do not meet their assumptions, they cannot guarantee identifiability. In fact, it is common for these conditions to be partially unmet. For instance, structural sparsity in [Zheng et al., 2022; Zheng & Zhang, 2023] is rarely satisfied for all latent variables, as verified by [Zheng & Zhang, 2023]. Similarly, [Zheng & Zhang, 2023; Kong et al., 2022] require at least $2n_A+1$ environments for the changing variables, which may be infeasible in certain cases. Therefore, in practical scenarios where assumptions may not hold universally, our approach offers flexible identifiability results through the strategy of learning by comparison, which these previous works do not address.
>
> 2. Our technical assumptions also differ in key ways:
>
> 	- [Zheng et al., 2022] assumes a sparsity structure between $\mathbf{z}$ and $\mathbf{x}$, disallowing dense Jacobians in the generative function $f$, while our approach places no constraints on $f$.
>
> 	- [Zheng & Zhang, 2023] also consider the structural sparsity on the mixing procedure between invariant latent variables and observed variables, while we allow arbitrary mixing structure. Moreover, they require $2n_A+1$ distinct environments for the changing latent variables, which we do not impose.
>
> 	- [Kong et al., 2022] addresses domain adaptation with a focus on content-style disentanglement, achieving component-wise identifiability for style variables but also requiring $2n_A+1$ distinct environments.
>
> 	Therefore, regarding assumptions on structures, [Zheng et al., 2022; Zheng & Zhang, 2023] assume a sparse structure in the generating function $f$ that maps latent variables $\mathbf{z}$ to observed variables $\mathbf{x}$ ($\mathbf{x} = f(\mathbf{z})$), whereas our approach does not impose any conditions on $f$. Instead, we focus solely on the structure between classes and concepts to determine which concepts can be reliably recovered. Furthermore, [Zheng & Zhang, 2023; Kong et al., 2022] require at least $2n_A+1$ distinct environments, an assumption we do not impose.
>
> **Summary:** Our work differs both in problem settings and technical conditions. Given the distinct objectives, we do not claim our theory is more general. Instead, our aim is to determine which concepts can be reliably identified with minimal assumptions in a broad setting. Our local identifiability theory can provide meaningful identifiability results with any pair of classes.

---

> > ### Author Response · Authors · 2024-11-15
> > **We are genuinely grateful for your insightful comments (2/2)**
> >
> > Following your valuable suggestions, we have added further discussion to emphasize these distinctions. Here are some examples:
> >
> > > “(Line 322) Different from various assumptions encouraging the sparsity of the structure in the literature (Rhodes & Lee, 2021; Moran et al., 2021; Zheng et al., 2022; Zheng & Zhang, 2023), our assumption only ensures necessary variability on the dependency structure and could also hold true with relatively dense connections. At the same time, we permit arbitrary structures between the class-dependent hidden concepts and the observed variables, while previous work has to assume a sparse structure on the generating process between latent and observed variables.”
> >
> > > “(Line 260) … Most latent variable identifiability works also face the same challenge dealing with partial assumption violation (Zheng et al., 2022; Kong et al., 2022; Zheng & Zhang, 2023, Hyvärinen et al., 2024). Unlike our local or even pair-wise identification strategy, these methods lack the flexibility to recover arbitrary parts of the hidden process in a localized manner.”
> >
> > > “(Line 328) Additionally, another line of work on latent variable models requires $2n_A + 1$ distinct domains or classes to achieve latent variable identifiability (e.g., (Hyvärinen & Morioka, 2017; Khemakhem et al., 2020a; Kong et al., 2022; Hyvärinen et al., 2024)), a condition we do not impose.”
> >
> >
> > Thank you once again for your insightful feedback, which has helped us contextualize our unique contributions. We hope these additions address your suggestions, and we would be grateful to hear if there are any further adjustments that could improve clarity or completeness.
> >
> > ---
> >
> > References:
> >
> > [1] Zheng et al., On the identifiability of nonlinear ICA: Sparsity and beyond, NeurIPS 2022
> >
> > [2] Zheng & Zhang, Generalizing nonlinear ICA beyond structural sparsity, NeurIPS 2023
> >
> > [3] Kong et al., Partial identifiability for domain adaptation, ICML 2022

---

> > ### Comment · Reviewer_NXKX · 2024-11-25
> >
> > Thank you for your response. Could you elaborate more on whether you think your experiments back up a conclusion that your methods (at least sometimes) compare favorably to those of  [Zheng et al., 2022; Zheng & Zhang, 2023; Kong et al. 2024]?
> >
> > Do you believe your experiments demonstrate situations in which concepts can be disentangled pair-wise for settings in which [Zheng et al., 2022; Zheng & Zhang, 2023; Kong et al. 2024] are unable to recover all the concepts?

---

> > > ### Author Response · Authors · 2024-11-26
> > > **Thanks so much for your further questions**
> > >
> > > Dear Reviewer NXKX,
> > >
> > > Thanks so much for your further questions. Yes, we believe our experiments demonstrate there are settings where the theories in [Zheng et al., 2022; Zheng & Zhang, 2023; Kong et al., 2022] do not support identifiability, but ours does. In addition to the **existing results**, we have also conducted **new experiments** in light of your great questions.
> > >
> > > Let’s start with existing results followed by additional experiments for comparison purposes. Please kindly find our detailed explanation below. It would be appreciated if you could let us know whether this addresses your concerns properly. We hope for your feedback and the opportunity to respond.
> > >
> > >
> > > **Existing Results:**
> > >
> > > In the **existing experiments (Figure 4)**, the data-generating process for model *Base* does not satisfy the sparsity condition in [Zheng et al., 2022; Zheng & Zhang, 2023], and does not meet the $2n+1$ domains requirement in [Kong et al., 2022]. As a result, we can clearly observe that the latent concepts can not be reliably recovered (Figure 4). In contrast, for model $Ours$, the addition of the structural diversity condition from our theory makes these latent concepts identifiable. This demonstrates situations where previous theories fail to ensure identifiability, but ours succeeds.
> > >
> > > It is worth noting that our structural diversity condition is a global formulation of learning by comparison across all pairs of classes, differing in both form and target from the structural sparsity condition on mixing function $\mathbf{x}=f(\mathbf{z})$ in [Zheng et al., 2022; Zheng & Zhang, 2023].
> > >
> > >
> > > **New Experiments:**
> > >
> > > To provide additional experimental evidence, we also conduct **new experiments** where assumptions in previous theories are partially violated (**Appendix C.2**). The datasets were generated under the following conditions:
> > >
> > > - *Base $(a)$*: Partially violates the structural sparsity assumption on the mixing structure between latent concepts and observed variables, as outlined in [Zheng et al., 2022; Zheng & Zhang, 2023]. Sparsity violations are introduced for a subset of concepts, with sizes randomly selected from 1 to $n/2$. The estimation method includes an $\ell_1$ regularization term on the mixing structure.
> > >
> > > - *Base $(b)$*: Partially violates the assumption of $2n+1$ domains in [Kong et al., 2022]. Latent concepts are generated from $n+1$ multivariate Gaussian distributions with zero mean and variances drawn from a uniform distribution over $[0.5, 3]$.
> > >
> > > - *Ours*: As $Ours$ in other experiments, it adheres to the proposed structural diversity condition. There are no constraints on the mixing structure between latent concepts and observed variables, but the structure between classes and concepts satisfies the required diversity. The estimation method includes an $\ell_1$ regularization term on the class-concept structure.
> > >
> > > For each model/data-generating process, we generate datasets with different numbers of concepts ($n \in \\{3,4,5,6\\}$). All results are from $10$ runs with different random seeds. The results (MCC) are shown in Figure 11 in the updated manuscript, which are summarized as follows:
> > >
> > > | Model   	| $n=3$     	| $n=4$     	| $n=5$     	| $n=6$     	|
> > > |-------------|-------------------|-------------------|-------------------|-------------------|
> > > | Base $(a)$  | 0.62±0.05     	| 0.53±0.11     	| 0.52±0.07     	| 0.44±0.11     	|
> > > | Base $(b)$  | 0.55±0.12     	| 0.48±0.16     	| 0.35±0.12     	| 0.43±0.18     	|
> > > | Ours    	| 0.86±0.05     	| 0.83±0.06     	| 0.88±0.03     	| 0.86±0.05     	|
> > >
> > > These results clearly demonstrate that partial violations of previous conditions also hinder the reliable recovery of latent concepts, highlighting more scenarios where prior conditions fail to provide identifiability guarantees but ours does.
> > >
> > >
> > >
> > > ---
> > >
> > > We sincerely appreciate the time and effort you have dedicated to reviewing our manuscript and providing valuable insights. With gratitude, we remain committed to addressing any remaining concerns. Please feel free to share any additional feedback you may have.
> > >
> > >
> > > With appreciation,
> > >
> > > Authors of Submission8461
> > >
> > > ---
> > >
> > > References:
> > >
> > > [1] Zheng et al., On the identifiability of nonlinear ICA: Sparsity and beyond, NeurIPS 2022
> > >
> > > [2] Zheng & Zhang, Generalizing nonlinear ICA beyond structural sparsity, NeurIPS 2023
> > >
> > > [3] Kong et al., Partial identifiability for domain adaptation, ICML 2022

---

> > > > ### Comment · Reviewer_NXKX · 2024-11-26
> > > >
> > > > Thank you for your response. I will update my rating to 6 (weak accept). I think higher rating would require experiments involving real-life datasets instead of synthetic data.

---

> ### Author Response · Authors · 2024-11-25
> **We sincerely appreciate your time and valuable feedback**
>
> Dear Reviewer NXKX,
>
> Thank you very much for your time and thoughtful suggestions. We hope we have adequately addressed your concerns. As the discussion period is nearing its conclusion, please feel free to share any additional feedback. We would be more than delighted to continue the discussion with you.
>
> Best wishes,
>
> Authors of Submission8461

---

> ### Author Response · Authors · 2024-11-28
> **We sincerely appreciate your suggestions; new real-world experiments have been conducted accordingly**
>
> Dear Reviewer NXKX,
>
> We sincerely appreciate your further feedback and suggestion that new real-world experiments could better support the claimed difference. In light of your great suggestions, we have conducted **new real-world experiments** on more complex scenarios. We would be truly grateful if you could let us know whether these updates adequately address your concerns. We look forward to your feedback and the opportunity to provide further clarifications if needed.
>
> Specifically, we performed experiments on the FFHQ dataset, which comprises $70,000$ human face images. The results, now highlighted in purple, are presented in **Figures 12**, **13**, and **14** in Appendix C.2 of the updated manuscript. These results reveal that **many concepts remain entangled** after estimation, indicating that the *assumptions required for full identifiability in previous theories may not be fully satisfied*. However, these class-related concepts are still identifiable as a group, *consistent with our theorem based on local or pairwise comparisons*. This provides clear real-world evidence of scenarios where **our theory offers a potential theoretical foundation, whereas previous guarantees fall short due to the lack of flexibility for local identification**.
>
> The detailed discussion is included as follows:
>
> > “Additional real-world experiments. To explore scenarios where not all concepts can be identified component-wise, we conduct additional real-world experiments on a more complex scenario, i.e., the FFHQ dataset (Karras et al., 2019). The dataset contains $70,000$ human face images, which is more complicated than the datasets in our other experiments. In addition to the estimation method introduced before, we incorporate a sparsity regularization ($\ell_1$ norm) on the Jacobian of the mixing function $f$, as required by (Zheng et al., 2022; Zheng & Zhang, 2023). Note that the identifiability theory in (Kong et al., 2022) does not require specific regularization during estimation if the task is not domain adaptation.”
>
> > “From Figs. 12, 13, and 14,  it is evident that some concepts remain entangled and cannot be fully recovered. For instance, for the class ``Age'', concepts like skin, eye, and face shape are all entangled together, suggesting that assumptions in (Zheng et al., 2022; Zheng & Zhang, 2023, Kong et al., 2022) for component-wise identifiability may not be fully satisfied in this scenario. However, these class-related concepts can still be identified as a group, consistent with our theorem based on local or pairwise comparisons. This suggests that, even in complex scenarios where prior theories fail to guarantee identifiability due to assumption violations, our alternative identifiability framework based on pairwise comparisons may still provide an alternative theoretical basis for recovering class-related concepts collectively, even if they remain entangled. This sheds light on the necessity of our alternative identifiability guarantees in some complicated real-world scenarios.”
>
>
> We believe these new experiments, alongside our existing results on four other real-world datasets, emphasize the significance of our theory in explaining the empirical success of concept learning across diverse settings. We are sincerely grateful for your constructive suggestions, which have significantly enhanced our manuscript. It has truly been a privilege to have you dedicate your time and effort to helping us improve our work.
>
> Thank you once again for your invaluable feedback.
>
> Many thanks,
>
> Authors of Submission8461

---

> > ### Author Response · Authors · 2024-12-02
> > **Did our new real-world experiments resolve the remaining concerns? Discussion ends today**
> >
> > Dear Reviewer NXKX,
> >
> > We are very grateful for your time and efforts devoted to improving our manuscript. Following your constructive suggestions (*"higher rating would require experiments involving real-life datasets instead of synthetic data"*), we have conducted and included **additional real-world experiments** to demonstrate the necessity and applicability of our theory in relevant scenarios.
> >
> > We would greatly appreciate it if you could kindly let us know whether your remaining concerns have been adequately addressed, or consider updating the rating to reflect these improvements.
> >
> > Thank you once again for your thoughtful feedback and contributions to the review process.
> >
> > With gratitude,
> >
> > Authors of Submission8461

---

### Meta-Review · Area_Chair_5c5n · 2024-12-21

**Metareview:**

The paper considers a problem of learning underlying concepts between pairs of classes. The tackled problem is interesting for ICLR, and the paper shows both theoretical results and evaluation on synthetic and some real-world data. The problem is essentially a causal discovery problem, but the setup and discussion is somewhat convoluted and unclear. The reviewers all had trouble getting the main ideas in the paper (as also evinced by the confidence scores). I suggest that the authors position the work in the context of causal estimation more clearly (as also suggested by reviewer Z5MD), and make the presentation and setup clearer. More details of the experimental setup would also be useful, perhaps even motivating the paper with one of the real-world examples in the introduction.

**Additional Comments On Reviewer Discussion:**

The authors made some attempts to clarify some aspects of the presentation and comparison to related work, which the reviewers have  mostly also incorporated in the review and scores. However, the issues regarding presentation, writing and broader contextualization do not seem fully addressed, and requires a more significant revision and re-evaluation.

---

### Decision · Program_Chairs · 2025-01-22

Reject